# VENUSX: UNLOCKING FINE-GRAINED FUNCTIONAL UNDERSTANDING OF PROTEINS

**Yang Tan**[1,2,4] *  **Wenrui Gou**[3] *  **Bozitao Zhong**[1]  **Huiqun Yu**[3]  **Liang Hong**[1,4] †
**Bingxin Zhou**[1] †
[1] Shanghai Jiao Tong University.
[2] Shanghai Innovation Institute.
[3] East China University of Science and Technology.
[4] Shanghai Matwings Technology Co., Ltd.

## ABSTRACT

Deep learning models have driven significant progress in predicting protein function and interactions at the protein level. While these advancements have been invaluable for many biological applications such as enzyme engineering and function annotation, a more detailed perspective is essential for understanding protein functional mechanisms and evaluating the biological knowledge captured by models. This study introduces VENUSX, the first benchmark designed to assess protein representation learning with a focus on fine-grained intra-protein functional understanding. VENUSX comprises three major task categories across six types of annotations, including residue-level binary classification, fragment-level multi-class classification, and pairwise functional similarity scoring for identifying critical active sites, binding sites, conserved sites, motifs, domains, and epitopes. The benchmark features over $878,000$ samples curated from major open-source databases such as InterPro, BioLiP, and SAbDab. By providing mixed-family and cross-family splits at three sequence identity thresholds, our benchmark enables a comprehensive assessment of model performance on both in-distribution and out-of-distribution scenarios. For baseline evaluation, we assess a diverse set of popular and open-source models, including pre-trained protein language models, sequence-structure hybrids, structure-based methods, and alignment-based techniques. Their performance is reported across all benchmark datasets and evaluation settings using multiple metrics, offering a thorough comparison and a strong foundation for future research. Our code, data, and a leaderboard are provided as open-source resources[1].

## 1 INTRODUCTION

Deep learning has significantly advanced the analysis of large-scale protein data, enabling efficient solutions to key inference tasks across sequence, structure, and function (Zhou et al., 2024a). Notable successes include structure prediction (Jumper et al., 2021; Abramson et al., 2024), sequence engineering (Lu et al., 2022; Zhou et al., 2024b; Tan et al., 2025b), and functional annotation (Yu et al., 2023; Wang et al., 2026). The rapid progress in this field is supported not only by the models' scientific and practical value, but also by the availability of high-quality benchmarks that define clear learning objectives and ensure fair, reproducible evaluation.

A wide range of datasets and evaluation protocols have been developed to facilitate model training and assessment, especially those centered on large-scale protein sequence and structure data (Orengo et al., 1997; Varadi et al., 2022; Consortium, 2025). While some benchmarks include functional annotations, they predominantly target protein-level properties, where the goal is to assign a single label to an entire protein or protein pair. Some representative tasks include function annotation (Xu et al., 2022; Tan et al., 2024b;a; YE et al., 2025; Li et al., 2025), protein–protein interaction prediction

---

*Equal contribution first authors.
†Corresponding authors (bingxin.zhou@sjtu.edu.cn; hongl3liang@sjtu.edu.cn).
[1]Code (VenusX Github), dataset (VenusX Huggingface), and leaderboard (VenusX Website).

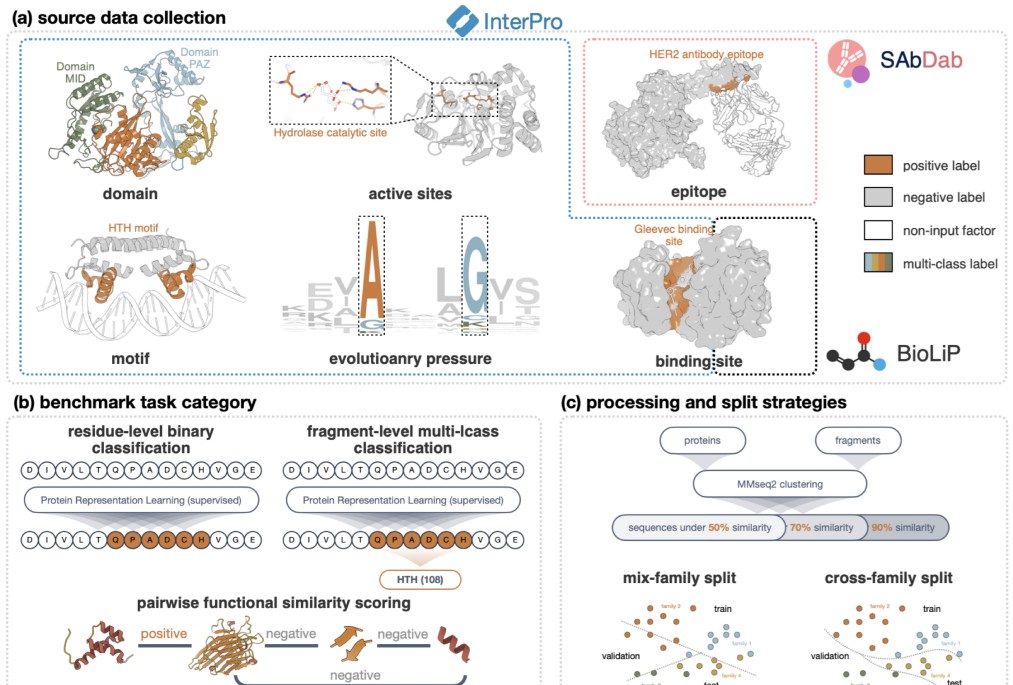

Figure 1: Overview of the VENUSX benchmark. (a) Six types of functional annotations collected from InterPro, BioLiP, and SAbDab (Section 2). (b) Three benchmark task categories: residue-level and fragment-level classification, and pairwise similarity scoring (Sections 3.1-3.3). (c) Sequence identity–based clustering and mix-family and cross-family data split strategies (Sections 3.4).

(Pan et al., 2010; Szklarczyk et al., 2019; Jankauskaitė et al., 2019; Liu et al., 2025), and protein fitness estimation (Gray et al., 2018; Riesselman et al., 2018; Notin et al., 2024; Zhang et al., 2025).

Despite the overwhelming focus on protein-level benchmarks, biological functions are often governed by specific subregions within proteins rather than the entire molecule. Global labels can obscure mechanistic details and may even lead models to rely on biologically implausible features for prediction. This increases the risk of overfitting to noise, reduces interpretability, and compromises accuracy in tasks where local features are critical, such as function annotation (Cagiada et al., 2023; Lee et al., 2007) and paratope design (Attique et al., 2023). As a result, **there is a growing demand for benchmarks that support supervision and evaluation at a fine-grained resolution**. It is essential not only for advancing functional understanding but also for systematically assessing how well learned representations capture biologically meaningful signals beyond sequence similarity.

We address this gap by introducing VENUSX, the first large-scale and biologically grounded benchmark for fine-grained protein understanding. VENUSX spans multiple subprotein levels—including residues, motifs, fragments, and domains—and is designed to evaluate model performance across three task categories: (1) **residue-level binary classification**, which assesses whether individual amino acids contribute critically to protein function, such as catalysis, ligand binding, evolutionary constraint, or domain boundaries; (2) **fragment-level multi-class classification**, which identifies functional subregions within a protein and assigns them to specific biological roles; and (3) **pairwise functional similarity scoring**, which matches functionally similar proteins or substructures without requiring explicit function labels.

The raw residue-level annotations are sourced from three high-quality databases: *InterPro* (Paysan-Lafosse et al., 2023), *BioLiP* (Yang et al., 2012), and *SAbDab* (Dunbar et al., 2014). We curate over 878,000 high-confidence samples and form diverse tasks across three categories of fine-grained functional prediction. To enable comprehensive evaluation of model fitness, robustness, and generalizability, we consider both label distribution and input similarity at the fragment and protein levels and define multiple evaluation setups with different partitioning strategies for training and testing.

We benchmark a broad spectrum of popular protein representative models to assess their effectiveness on VENUSX. These include pre-trained protein language models (Rives et al., 2021; Elnaggar et al., 2021; 2023; Heinzinger et al., 2023; Lin et al., 2023; Hamamsy et al., 2024), sequence-structure hybrid models (Su et al., 2023; Tan et al., 2025c), inverse folding models (Yang et al., 2023; Hsu et al., 2022), structure-based geometric networks (Jing et al., 2021), and traditional alignment-based methods (Altschul et al., 1990; Zhang & Skolnick, 2005; Van Kempen et al., 2024). We observe substantial variation in performance across annotation types, sequence identity thresholds, and task formulations. These findings reveal that strong performance on conventional global protein-level tasks does not necessarily translate to fine-grained functional understanding. The results further suggest that many current models rely heavily on global or distributional cues, rather than capturing precise, localized biological signals. These limitations highlight the need for future model designs that are better aligned with the demands of fine-grained benchmarks, which emphasize robustness, generalization across protein families, and biological interpretability.

In summary, while prior benchmarks have focused on protein-level properties, VENUSX establishes the first comprehensive, multi-level benchmark for fine-grained protein understanding by introducing a suite of tasks at the residue, fragment, and domain levels. It offers biologically meaningful evaluation dimensions for future protein models and enables systematic assessment of their ability to capture true biological knowledge. All datasets, task definitions, evaluation protocols, and baseline leaderboards are publicly available to support followup research in the future.

## 2 DATA COLLECTION AND CURATION

We collect residue- and fragment-level annotations from **InterPro** (Paysan-Lafosse et al., 2023), **BioLiP** (Yang et al., 2012), and **SAbDab** (Dunbar et al., 2014), followed by thorough cleaning, redundancy removal, identity-based clustering, and alignment with structure and annotations.

### 2.1 INTERPRO: FUNCTIONAL ANNOTATIONS ACROSS DIVERSE PROTEIN FAMILIES

We use InterPro (Paysan-Lafosse et al., 2023) to collect residue-level entries from various annotation categories, such as active sites (Act), binding sites (Bind), conserved sites (Evo), functional motifs, and domains, detailed definitions for each of these annotation types are provided in Appendix Section A.1. For each category, metadata are retrieved from InterPro Website [2] in `.json` format, including InterPro family identifiers, associated Gene Ontology (GO) terms, and annotated residue positions. The raw data of UniProt identifiers and their corresponding functional annotations are downloaded by VenusFactory (Tan et al., 2025a). Canonical protein sequences are obtained from UniProt (Consortium, 2025), and predicted structures are retrieved from the AlphaFold Protein Structure Database (Varadi et al., 2022), retaining only entries with available structural models.

To ensure non-redundancy, we remove additional entries with identical annotated fragments. If a protein contains multiple distinct fragments annotated with the same function, the annotations are consolidated into a single entry. These fragments, along with their aligned sequence-structure pairs, form the basis for the various downstream benchmark tasks.

### 2.2 BIOLIP: EXPERIMENTALLY-DERIVED LIGAND BINDING SITES

We incorporate additional residue-level binding site annotations from BioLiP (Yang et al., 2012), a curated resource of protein–ligand interactions derived from experimentally resolved complexes in the Protein Data Bank (PDB) (Burley et al., 2019). Binding residues are identified using a distance-based criterion. A residue is labeled as part of the binding site if any of its atoms lie within the sum of the Van der Waals radii of the interacting atom pair plus a $0.5$ Å empirical margin. This approach captures steric interactions and enables robust identification of physiologically relevant binding interfaces.

For each entry, we extracted the complete receptor sequence, its PDB chain identifier, the annotated binding site residues, and the ligand identity specified by its Chemical Component Dictionary (CCD) code. Sequence and coordinate information were parsed directly from the corresponding PDB files, and ligand annotations were stringently curated to retain only biologically relevant molecules. This

---

[2]`https://www.ebi.ac.uk/interpro/`

Table 1: Summary of the 7 residue-level classification tasks. For sequence length, number of positive residues, and proportion of positives, averages are shown with standard deviations in parentheses.

| Target | Description | # Proteins | Seq. Len. | # Positive | % Positive | Cross | Mix |
|--------|-------------|-----------:|-----------|-----------:|-----------:|:-----:|:---:|
| **Act** | active sites (InterPro) | $9,667$ | $482.5_{(349.0)}$ | $16.7_{(7.0)}$ | $4.6_{(2.8)}$ | ✓ | ✓ |
| **BindI** | binding sites (InterPro) | $8,959$ | $486.5_{(339.8)}$ | $24.5_{(20.9)}$ | $7.9_{(7.3)}$ | ✓ | ✓ |
| **BindB** | binding sites (BioLiP) | $115,505$ | $348.7_{(272.9)}$ | $9.4_{(9.6)}$ | $4.1_{(5.7)}$ | | ✓ |
| **Evo** | evolutionary pressure (InterPro) | $59,948$ | $365.9_{(290.8)}$ | $23.4_{(13.1)}$ | $10.4_{(9.2)}$ | ✓ | ✓ |
| **Motif** | motif (InterPro) | $10,271$ | $595.2_{(383.5)}$ | $78.0_{(73.4)}$ | $20.2_{(22.8)}$ | ✓ | ✓ |
| **Dom** | domain (InterPro) | $595,454$ | $537.5_{(373.2)}$ | $169.2_{(117.2)}$ | $40.3_{(26.1)}$ | ✓ | ✓ |
| **Epi** | epitope (SAbDab) | $5,370$ | $374.9_{(291.3)}$ | $24.8_{(10.5)}$ | $10.6_{(9.2)}$ | | ✓ |

curated dataset augments the InterPro-derived entries with high-resolution protein–ligand binding-site pairs, providing a biologically meaningful benchmark for training and evaluating deep learning models on protein-ligand complexes.

## 2.3 SAbDab: Antibody-independent Epitope Prediction

This benchmark introduces a task for antibody-independent epitope prediction, a well-established and critical challenge in computational biology that is pivotal for accelerating the design of next-generation therapeutic antibodies and vaccines (Cia et al., 2023; Zeng et al., 2023; Carroll et al., 2024). The task aims to identify an antigen's intrinsically bindable or druggable regions, which serve as high-potential targets for therapeutic antibodies. This approach is motivated by the observation that functional antibodies often target specific, high-affinity regions rather than the entire exposed protein surface (Hastie et al., 2021). To construct a dataset for this task, we extract epitope annotations from antibody-antigen complexes curated in SAbDab (Dunbar et al., 2014), restricting our analysis to entries where the antigen is a protein. Metadata are sourced from curated `.tsv` files from SAbDab Website[3], including antibody heavy and light chains, as well as the associated antigen chains. Structures are parsed using BioPython (Cock et al., 2009). We only retain standard residues with defined $C_\alpha$ coordinates. An antigen residue is considered part of the epitope if the Euclidean distance between its $C_\alpha$ atom and any antibody $C_\alpha$ atom is less than 10 Å. This geometric criterion captures both continuous epitopes (adjacent in sequence) and conformational epitopes (spatially clustered but sequence-distant).

We treat antigens with identified epitope residues from this procedure as valid entries. For each of them, we extract the complete amino acid sequence, the epitope residue indices (start from 0), and associated structure files. Entries with non-protein antigens, missing chain information, or structural inconsistencies are excluded. The resulting dataset offers structure-derived epitope labels aligned with sequence data to support targeted development and evaluation of immune-related protein models.

## 3 Benchmark Tasks

### 3.1 Residue-Level Binary Classification

**Task Description**  The first category of tasks focuses on identifying functionally important residues within a protein. Each task is framed as a binary classification problem at the residue level, where positions are labeled as either functionally relevant (positive) or irrelevant (negative). Related functions involve catalysis, binding, or other biological processes (Section 2). Table 1 summarizes all 7 subtasks. Unlike conventional protein-level function prediction, these tasks assess whether models can detect critical residues independent of global function or family context. They encourage models to go beyond coarse representations and capture residue-level signals that may support model analysis in terms of interpretability or explanability. In practice, accurate residue-level predictions can aid enzyme engineering, mutational effect analysis, and active-site redesign.

---

[3]`https://opig.stats.ox.ac.uk/webapps/sabdab-sabpred/sabdab/search/`

Table 2: Summary of the 5 fragment-level multi-class classification tasks.

| Target | # Fragments | Seq. Len. | # Class |
|--------|------------:|-----------|--------:|
| **Act** | 9,767 | 18.7 $_{(7.0)}$ | 132 |
| **BindI** | 10,562 | 26.6 $_{(21.7)}$ | 76 |
| **Evo** | 66,916 | 25.5 $_{(13.3)}$ | 740 |
| **Motif** | 13,244 | 80.23 $_{(73.8)}$ | 454 |
| **Dom** | 656,669 | 171.3 $_{(117.4)}$ | 13,459 |

Table 3: Summary of the five pairwise similarity scoring tasks (in millions).

| Target | Protein | | Fragment | |
|--------|-----------:|-----------:|-----------:|-----------:|
| | # Positive | # Negative | # Positive | # Negative |
| **Act** | 1.3 | 45.4 | 1.3 | 46.4 |
| **BindI** | 3.5 | 36.6 | 5.0 | 50.8 |
| **Evo** | 7.7 | 1,789.1 | 10.0 | 2,228.9 |
| **Motif** | 2.4 | 50.3 | 6.0 | 81.7 |
| **Dom** | 217.7 | 177,064.8 | 346.0 | 215,260.7 |

**Labeling Strategy**  For these residue-level classification tasks, we adopt a specific labeling approach. If a protein contains multiple, spatially distinct sites annotated with the same function (e.g., several separate binding sites), all residues belonging to any of these sites are labeled as positive (1), while all other residues are labeled as negative (0).

**Evaluation Metrics**  These tasks exhibit strong class imbalance, as most residues are non-functional (see "% Positive" in Table 1). To better assess model performance on the minority class, we prioritize metrics that emphasize positive predictions. Specifically, we report class-specific precision, recall, and F1 score for the positive class, along with the area under the precision–recall curve (AUPR).

## 3.2 FRAGMENT-LEVEL MULTI-CLASS CLASSIFICATION

**Task Description**  The second category task evaluates a model's ability to classify pre-identified functional regions. Unlike remote homology detection (Rao et al., 2019), which typically uses a full-length protein as input, this task provides only the annotated fragment sequence. The objective is to assign this fragment to its corresponding InterPro family (see Table 2). This reflects a practical two-stage inference pipeline that first locates functionally relevant regions, then assigns them to functional families. Since proteins often contain multiple functional fragments, the task also supports multi-label (InterPro family) annotations and encourages models to capture compositional functionality. It also bridges residue-level predictions with interpretable biological labels grounded in existing ontologies. Crucially, inputs for this task are continuous sequence motifs (e.g., signature sequences defined by InterPro). We do not artificially concatenate non-consecutive residues. Consequently, tasks defined solely by 3D spatial proximity without conserved 1D motifs (e.g., BioLiP binding sites, SAbDab epitopes) are excluded from this category and handled exclusively in residue-level tasks.

**Labeling Strategy**  The label for each functional fragment is its corresponding InterPro family ID, as curated from the source database. Since a single protein region can occasionally be assigned to multiple overlapping InterPro families, the task is formulated as a multi-label classification problem, though most instances have a single label.

**Evaluation Metrics**  These tasks assign each functional fragment to its corresponding InterPro family, which may include hundreds to tens of thousands of distinct classes (see "# Class" in Table 2). We report accuracy (ACC), macro-averaged precision, recall, F1 score, and Matthews correlation coefficient (MCC). We take ACC and macro-F1 as the primary metrics to reflect both overall correctness and class-balanced performance.

## 3.3 PAIRWISE FUNCTIONAL SIMILARITY SCORING

**Task Description**  This third category of tasks evaluates how well learned representations can capture functional similarity in a zero-shot, unsupervised setting. Given a pair of inputs (either proteins or fragments), the goal is for a model to produce a similarity score reflecting their functional relatedness. This task provides a retrieval-style evaluation of representation quality, particularly for identifying subtle but biologically relevant similarities. It has important practical value in applications such as enzyme mining, remote homolog detection, and functional clustering in metagenomic datasets. A full dataset summary is provided in Table 3.

**Labeling Strategy**   The ground truth for this task is defined by InterPro family membership. A pair of proteins or fragments is considered a positive sample if both members are annotated with the same InterPro family ID. Conversely, a pair is considered a negative sample if they belong to different InterPro families.

**Evaluation Metrics**   Pairwise similarity scoring tasks are evaluated using the area under the ROC curve (AUC). We use cosine similarity between protein representations as the similarity score for embedding-based methods. For alignment-based methods that explicitly compute sequence or structure alignments, we employ the negative logarithm of the E-value (*e.g.*, for FOLDSEEK (Van Kempen et al., 2024) and BLAST (Altschul et al., 1990)) or the bi-directional average TM-score (*e.g.*, for TM-ALIGN (Zhang & Skolnick, 2005)).

### 3.4 PARTITIONING PROTOCOL FOR TRAINING AND EVALUATION

**Classification Tasks**   We evaluate both in-distribution and out-of-distribution prediction performance. To this end, we construct mix-family and cross-family data splits for both residue-level and fragment-level tasks. (1) Mix-family splits assess in-distribution generalization by randomly partitioning proteins (or fragments) into training, validation, and test sets in an 8:1:1 ratio, without considering family assignments. (2) Cross-family splits evaluate out-of-distribution generalization by assigning entire InterPro families to training, validation, and test sets in the same 8:1:1 ratio. For both strategies, we first apply MMseqs2 clustering (Steinegger & Söding, 2017) at 50%, 70%, and 90% sequence identity thresholds to reduce redundancy before splitting. Note that, due to limitations in available data and family annotations across the original source databases, mix-family splits on proteins are applied to datasets from all three sources. In contrast, fragment-level splits and cross-family splits based on family identity are applied only to InterPro-sourced datasets. Detailed availability and train/validation/test set statistics are provided in Tables 9–10 in Appendix Section A.5.

**Similarity Scoring Tasks**   As the third task category of pairwise functional similarity scoring does not involve model supervision, we do not perform data partitioning. Instead, following a similar strategy to (Tan et al., 2024c), we uniformly subsample a set of positive and negative pairs from the complete dataset for evaluation. This random subsampling is necessary due to the combinatorially large number of possible protein pairs in the similarity scoring task (for instance, see Table 3). Specifically, for both pairing tasks on fragments and proteins, we randomly sample $10,000$ positive pairs (*i.e.*, proteins from the same InterPro family) and $10,000$ negative pairs (*i.e.*, proteins from different InterPro families) as one evaluation dataset. We repeat the procedure using three different random seeds, and the final performance scores are averaged across the three repetitions.

### 3.5 NAMING PROTOCOL FOR BENCHMARK DATASETS

Following the construction options introduced in Sections 3.1-3.4, VENUSX includes a total of 56 datasets. For clarity, each dataset is named by `VENUSX_[category]_[target]_[split]`, where each part denotes the task category, prediction target, and data split strategy.

- `[category]` refers to the task category, with three choices of `Res`, `Frag`, and `Pair` to represent Residue-level tasks, fragment-level tasks, and pairwise scoring tasks.

- `[target]` represents the 7 cases of targets, including `Act` (active sites), `BindI` (binding sites from InterPro), `BindB` (binding sites from BioLiP), `Evo` (evolutionary pressure), `Motif` (functional motif), `Dom` (functional domain), and `Epi` (epitope sites).

- `[split]` denotes the partitioning strategies, with `X` for cross-family and `M` for mix-family splits, `P` and `F` indicating protein or fragment, and the final number representing the clustering threshold.

For instance: (1) `VENUSX_Res_BindB_X` is a residue-level binary classification task that predicts binding sites (from BioLiP), using a cross-family split. (2) `VENUSX_Frag_Act_MF90` denotes a fragment-level multi-class classification task targeting active sites, with a mix-family split on fragment entries clustered at $90\%$ identity. This benchmark is designed for community expansion, and we welcome contributions. Upon acceptance, a public repository will be made available for contributions via Pull Request. The primary data requirement is a format including a `seq_full`

Table 4: Residue-level classification performance across datasets and data splits. "MF50" and "MP50" refer to mixed-family splits with 50% sequence identity filtering applied to fragments and proteins, respectively. **Top-1**, Top-2, and *Top-3* results for each target dataset are highlighted, respectively. Models are grouped by input modality. AUPR scores for each task are reported, and detailed results are provided in Tables 19-21 of the Appendix Section F.

| Target | Split | Sequence-only | | | | Sequence-Structure | | | Structure-only |
|---|---|---|---|---|---|---|---|---|---|
| | | ESM2-T30 | ESM2-T33 | PROTBERT | ANKH-BASE | SAPROT-35M | SAPROT-650M | PROTSSN | GVP-GNN |
| **Act** | MF50 | 0.855 | *0.852* | 0.764 | **0.873** | 0.688 | 0.745 | 0.465 | 0.523 |
| | MP50 | 0.932 | 0.955 | 0.895 | **0.960** | 0.905 | *0.945* | 0.917 | 0.898 |
| | Cross | 0.143 | 0.143 | 0.131 | 0.166 | 0.114 | **0.185** | *0.156* | 0.101 |
| **BindI** | MF50 | **0.912** | *0.904* | 0.857 | 0.907 | 0.807 | 0.838 | 0.801 | 0.611 |
| | MP50 | *0.963* | **0.971** | 0.926 | 0.970 | 0.927 | 0.960 | 0.907 | 0.883 |
| | Cross | 0.133 | *0.159* | 0.112 | 0.145 | **0.230** | 0.182 | 0.182 | 0.040 |
| **Evo** | MF50 | *0.862* | **0.899** | 0.771 | 0.895 | 0.724 | 0.734 | 0.715 | 0.342 |
| | MP50 | 0.897 | 0.926 | 0.803 | **0.932** | 0.775 | *0.912* | 0.895 | 0.792 |
| | Cross | 0.235 | *0.262* | 0.243 | **0.275** | 0.272 | 0.274 | 0.227 | 0.101 |
| **Motif** | MF50 | *0.855* | 0.874 | 0.779 | **0.884** | 0.767 | 0.802 | 0.716 | 0.661 |
| | MP50 | *0.850* | 0.857 | 0.796 | **0.870** | 0.784 | 0.841 | 0.765 | 0.736 |
| | Cross | *0.433* | **0.456** | 0.348 | 0.394 | 0.408 | 0.441 | 0.390 | 0.329 |
| **Dom** | MF50 | 0.634 | 0.666 | 0.591 | **0.673** | 0.574 | *0.642* | – | 0.560 |
| | MP50 | 0.645 | *0.657* | 0.592 | **0.665** | 0.584 | 0.640 | – | 0.557 |
| | Cross | 0.470 | 0.506 | *0.508* | 0.449 | 0.525 | **0.564** | – | 0.468 |
| **BindP** | MP50 | *0.409* | **0.446** | 0.340 | 0.421 | 0.374 | 0.388 | – | 0.301 |
| | MP70 | *0.465* | **0.494** | 0.410 | 0.487 | 0.330 | 0.389 | – | 0.337 |
| | MP90 | *0.496* | **0.535** | 0.466 | 0.527 | 0.354 | 0.411 | – | 0.374 |
| **Epi** | MP50 | *0.186* | 0.174 | 0.169 | 0.167 | 0.182 | 0.195 | **0.196** | 0.118 |
| | MP70 | 0.184 | *0.202* | 0.177 | 0.215 | 0.194 | **0.237** | 0.200 | 0.139 |
| | MP90 | *0.277* | 0.290 | 0.266 | *0.270* | 0.256 | **0.308** | 0.274 | 0.196 |

field for the full-length sequence and a `label` field for the corresponding residue-level annotation array, with detailed instructions provided in the supplementary materials.

## 4 EXPERIMENTS

### 4.1 EXPERIMENTAL SETUP

**Model Setup** For both residue- and fragment-level classification tasks, pretrained sequence-based models (e.g., ESM2 (Lin et al., 2023), PROTBERT (Elnaggar et al., 2021)) and sequence–structure models (SAPROT (Su et al., 2023), PROTSSN (Tan et al., 2025c)) are used as frozen feature extractors. This protocol isolates the intrinsic quality of pre-trained representations from fine-tuning confounders and ensures computational accessibility on this large-scale benchmark. In contrast, the structure-based GVP-GNN (Jing et al., 2021) is trained from scratch with all parameters updated, this approach aligns with standard evaluation protocols for geometric GNNs (Jamasb et al., 2024) and ensures fairness in computational scale. For residue-level tasks, the encoders output embeddings of each residue, which are passed through two linear layers with ReLU activation and dropout. For fragment-level classification, mean pooling is applied to obtain fragment representations for InterPro family prediction. In pair-level similarity evaluation, full-length sequences or fragments are encoded, mean-pooled, and compared via similarity metrics to assess family-level relationships. Parameter statistics for all models are provided in Table 13 in Appendix Section B.

**Training Setup** For all tasks, full-length protein sequences are truncated to a maximum of 1022 residues. For sequence-structure models, 3D structure files were truncated in parallel to include only the atoms of the retained residues, ensuring a perfect residue-to-coordinate mapping. Fragments are capped at 128 residues for Act, BindI, Evo, and Motif, and at 512 residues for Dom. All models are trained with a fixed random seed of 3407 to ensure reproducibility. Optimization is performed using AdamW (Loshchilov et al., 2017) with a learning rate of 0.001 and an effective batch size of 128 via gradient accumulation. Training proceeds for up to 100 epochs, with early stopping triggered if validation performance does not improve for 10 epochs. For residue-level and fragment-level classification tasks, AUPR and accuracy on the validation set are used as early stopping criteria, respectively. All experiments are conducted on 16 NVIDIA RTX 4090D GPUs and 192 Intel(R) Xeon(R) Gold 6248R CPUs with 2 TB of memory for 45 days.

Table 5: Fragment-level classification performance across InterPro datasets and data splits under 50% sequence identity. **Top-1**, Top-2, and *Top-3* results for each metric are highlighted, respectively. Detailed results are provided in Table 22 of Appendix Section F

| Target | Metric | Sequence-only | | | | Sequence-Structure | | | Structure-only |
|---|---|---|---|---|---|---|---|---|---|
| | | ESM2-T30 | ESM2-T33 | PROTBERT | ANKH-BASE | SAPROT-35M | SAPROT-650M | PROTSSN | GVP-GNN |
| **Act** | ACC | 0.819 | 0.814 | 0.736 | 0.824 | **0.928** | **0.928** | *0.891* | 0.907 |
| | Macro-F1 | 0.647 | 0.605 | 0.609 | 0.647 | *0.807* | 0.825 | 0.764 | **0.906** |
| **BindI** | ACC | 0.937 | 0.934 | 0.927 | 0.920 | 0.976 | **0.986** | *0.972* | *0.972* |
| | Macro-F1 | *0.913* | 0.753 | 0.790 | 0.718 | 0.809 | **0.957** | 0.931 | 0.884 |
| **Evo** | ACC | 0.853 | 0.841 | 0.828 | 0.866 | 0.939 | **0.950** | *0.915* | 0.914 |
| | Macro-F1 | 0.667 | 0.669 | 0.627 | 0.716 | 0.849 | **0.863** | *0.793* | 0.757 |
| **Motif** | ACC | 0.884 | *0.906* | 0.884 | 0.901 | 0.901 | **0.927** | 0.914 | 0.807 |
| | Macro-F1 | 0.457 | *0.542* | 0.452 | 0.499 | 0.504 | 0.552 | **0.556** | 0.370 |

Table 6: AUC (%) of baseline models on InterPro family alignment under two evaluation settings: **F50** (fragment-level inputs with 50% sequence identity filtering) and **P50** (full-sequence inputs with 50% identity). Models are grouped by modality. Cell colors indicate ranking: ■ Top-1, ■ Top-2, ■ Top-3, ■ Top-4. Standard deviation over three folds is shown in parentheses.

| Model Information | | Act | | BindI | | Evo | | Motif | | Dom | |
|---|---|---|---|---|---|---|---|---|---|---|---|
| Name | Version | F50 | P50 | F50 | P50 | F50 | P50 | F50 | P50 | F50 | P50 |
| **Alignment-based Methods** | | | | | | | | | | | |
| FOLDSEEK | 3Di | 96.0(0.1) | 96.5(0.2) | 92.6(0.2) | 80.6(0.2) | 88.3(0.1) | 99.0(0.1) | 74.8(0.2) | 64.9(0.1) | – | – |
| | 3Di-AA | 96.1(0.1) | 96.5(0.2) | 92.6(0.2) | 80.1(0.2) | 88.4(0.2) | 99.0(0.1) | 74.7(0.2) | 64.7(0.2) | – | – |
| TM-ALIGN | mean | 94.6(0.0) | – | 90.1(0.1) | – | 67.7(0.1) | – | 76.6(0.0) | – | – | – |
| BLAST | – | 52.9(0.2) | 71.7(0.1) | 52.4(0.1) | 51.1(0.0) | 54.0(0.3) | – | 49.9(0.1) | 56.2(0.3) | – | – |
| **Sequence-only Encoder Methods** | | | | | | | | | | | |
| ESM2 | t30 | 69.4(0.5) | 69.2(0.2) | 77.6(0.4) | 65.5(0.2) | 52.4(0.5) | 87.5(0.2) | 84.3(0.5) | 68.2(0.3) | 78.0(0.2) | 77.4(0.0) |
| | t33 | 50.2(0.5) | 70.0(0.3) | 73.0(0.4) | 62.3(0.3) | 49.3(0.5) | 89.0(0.2) | 92.1(0.3) | 66.1(0.1) | 62.2(0.2) | 66.4(0.1) |
| | t36 | 65.8(0.3) | 72.9(0.2) | 71.3(0.3) | 67.6(0.4) | 63.9(0.1) | 92.1(0.0) | 90.1(0.3) | 70.0(0.1) | 66.5(0.2) | 66.7(0.0) |
| ESM-1B | t33 | 67.6(0.2) | 73.8(0.2) | 84.5(0.2) | 69.8(0.2) | 57.0(0.5) | 88.4(0.3) | 87.2(0.3) | 58.4(0.4) | 89.2(0.2) | 74.7(0.2) |
| PROTBERT | bfd | 71.4(0.5) | 68.7(0.3) | 84.9(0.4) | 66.8(0.1) | 54.6(0.4) | 84.2(0.3) | 85.1(0.2) | 68.2(0.3) | 85.3(0.2) | 77.9(0.3) |
| **Sequence-only Encoder-Decoder Methods** | | | | | | | | | | | |
| PROTT5 | xl_uniref50 | 91.8(0.1) | 78.1(0.2) | 98.5(0.1) | 77.1(0.1) | 71.0(0.2) | 95.6(0.1) | 98.2(0.0) | 67.6(0.3) | 98.5(0.1) | 85.1(0.1) |
| ANKH | base | 69.6(0.5) | 90.4(0.2) | 88.9(0.2) | 91.8(0.2) | 63.9(0.4) | 98.9(0.1) | 86.7(0.2) | 69.7(0.3) | 97.6(0.1) | 88.5(0.1) |
| **Sequence-structure Methods** | | | | | | | | | | | |
| SAPROT | 35M_AF2 | 95.8(0.0) | 74.6(0.2) | 94.3(0.1) | 71.9(0.2) | 61.9(0.5) | 92.7(0.2) | 85.3(0.1) | 66.6(0.2) | 96.0(0.1) | 78.8(0.3) |
| | 650M_PDB | 82.8(0.2) | 68.2(0.1) | 98.1(0.1) | 71.1(0.1) | 62.6(0.5) | 93.8(0.1) | 98.9(0.0) | 68.3(0.3) | 91.7(0.1) | 76.1(0.2) |
| PROTSSN | k20_h512 | 79.1(0.3) | 64.8(0.2) | 88.4(0.4) | 61.2(0.4) | 60.9(0.4) | 86.2(0.1) | 72.4(0.3) | 64.0(0.2) | 82.9(0.2) | 69.4(0.0) |
| ESM-IF | – | 96.5(0.1) | 70.2(0.2) | 95.0(0.1) | 65.6(0.1) | 61.3(0.3) | 90.6(0.4) | 80.4(0.2) | 66.0(0.2) | 97.1(0.2) | 70.5(0.2) |
| MIF-ST | – | 65.9(0.6) | 65.9(0.3) | 86.1(0.1) | 59.2(0.3) | 61.3(0.4) | 80.3(0.2) | 50.2(0.4) | 66.3(0.6) | 78.6(0.2) | 66.7(0.0) |
| TM-VEC | swiss_large | 93.6(0.2) | 89.9(0.2) | 98.6(0.0) | 82.4(0.0) | 67.4(0.2) | 96.2(0.1) | 99.4(0.0) | 71.7(0.3) | 98.2(0.1) | 59.9(0.2) |
| PROSTT5 | AA2fold | 90.8(0.1) | 80.7(0.3) | 99.5(0.0) | 79.2(0.0) | 55.6(0.5) | 98.2(0.0) | 98.5(0.0) | 69.8(0.2) | 98.5(0.1) | 79.3(0.2) |

## 4.2 EVALUATED METHODS

The summary of all baseline models, including architecture type, version, task scope, parameter count, and implementation source, is presented in Table 13 in Appendix.

**Classification Baselines** For the residue-level and fragment-level classification tasks, we evaluate a set of pretrained models spanning diverse architectures. These include sequence-based language models such as ESM2 (Lin et al., 2023), PROTBERT (Elnaggar et al., 2021), and ANKH (Elnaggar et al., 2023); sequence–structure hybrid models such as SAPROT (Su et al., 2023) and PROTSSN (Tan et al., 2025c); and a structure-only geometric network, GVP-GNN (Jing et al., 2021).

**Similarity Scoring Baselines** For the pair-wise similarity evaluation task, we extract mean embeddings from pretrained language models, including ESM2 (Lin et al., 2023), PROTBERT (Elnaggar et al., 2021), ESM-1B (Rives et al., 2021), PROTT5 (Elnaggar et al., 2021), ANKH (Elnaggar et al., 2023), TM-VEC (Hamamsy et al., 2024), and two inverse folding models, MIS-ST (Yang et al., 2023) and ESM-IF1 (Hsu et al., 2022). For traditional alignment-based methods, we include BLAST (Altschul et al., 1990) (sequence alignment), TM-ALIGN (Zhang & Skolnick, 2005) (structure align-

ment), and FOLDSEEK (Van Kempen et al., 2024), which supports both structure-only (3Di) and joint sequence-structure (3Di-AA) comparisons.

## 4.3 RESIDUE-LEVEL BINARY CALSSIFICATION

Table 4 lists AUPR on seven residue-annotation benchmarks under mixed-family (in-distribution) and cross-family (out-of-distribution) splits. Some experiments are omitted due to missing structural inputs or prohibitive computational costs. Key observations are:

- **Language models perform strongly on in-distribution splits.** ANKH-BASE attains the highest AUPR on 7/15 InterPro Mix tasks, while ESM2-T33 dominates the *BindP* and *Epi* mixes, confirming that sequence is sufficient when test proteins remain close to the training set.

- **Sequence–structure models generalize better for unseen families.** SAPROT-650M achieves the best or second-best performance across all InterPro Cross splits, and shows a notable advantage in domain-level classification (e.g., +5.6% AUPR over PROTBERT).

- **Cross-family residue prediction remains highly challenging.** On *Act* and *BindI* the best AUPR plummets by 70–80% in Cross, whereas *Dom* drops by $< 10\%$, suggesting that catalytic and binding residues are harder to extrapolate than domain-wide patterns.

- **Dataset properties strongly affect difficulty.** Mix splits from InterPro are relatively well-structured and easier to predict. In contrast, *Epi* remains extremely difficult—no model achieves an AUPR above 0.3 across all sequence identity levels.

## 4.4 FRAGMENT-LEVEL MULTI-CLASS CLASSIFICATION

Table 5 reports ACC and Macro-F1 on four InterPro targets (Act, BindI, Evo, Motif) with 50% sequence-identity filtering.

- **Sequence-structure models are consistently superior.** SAPROT-650M or PROTSSN ranks first on 7/8 metrics; e.g. SAPROT-650M outperforms the strongest protein language model (ESM2-T33) on *Act* by +11.4% ACC and +22% Macro-F1.

- **Structure-only models show strong task-specific performance.** GVP-GNN matches alignment-aware models on *Act* and *BindI* (Macro-F1 = 0.906 / 0.884) but lags on *Motif*, indicating pure structure is task-dependent.

- **Sequence models are more sensitive to class imbalance.** While ACC exceeds 80%, their Macro-F1 is 15–20% lower; sequence-structure models cut this gap to ∼10%, showing better robustness to skewed label distributions.

## 4.5 PAIRWISE FUNCTIONAL SIMILARITY SCORING

Table 6 reports AUC (%) for family-level alignment under two low-identity conditions—**F50** (fragment inputs clustered at 50% identity) and **P50** (protein inputs at 50% identity). Because exhaustive structural alignment (e.g., TM-ALIGN) is prohibitively expensive at this scale, several entries are left blank; baseline models description and full command lines are given in Appendix Section B.2.

- **Structure-based aligners remain the gold standard.** FOLDSEEK delivers near-perfect performance, topping 3/8 settings and peaking at 99.0% AUC on *Evo_P50*. TM-ALIGN is likewise competitive when evaluated, whereas sequence-only BLAST trails by $> 40\%$ on every task, underscoring the advantage of structural information.

- **Large encoder–decoder models close much of the gap.** PROTT5 attains 98.5% on *BindI_F50* and 98.2% on *Motif_F50*, outperforming all pure sequence encoders (ESM2, ESM-1B, PROTBERT) by 7–20% and surpassing TM-ALIGN on three fragment settings. Ankh shows similar strength on full-sequence inputs (*Act_P50*: 90.4% vs. 69–74% for vanilla encoders).

- **Sequence–structure hybrids are highly alignment-aware.** TM-VEC reaches top-2 ranking in 5 of 10 cells (e.g., 99.4% on *Motif_P50*; 98.2% on *Dom_P50*), while SAPROT-650M attains 98.1% on *BindI_P50*. These results indicate that injecting structural inductive bias enables LM embeddings to rival specialised aligners at a fraction of the computational cost.

## 5   RELATED WORK

**Protein-wise Tasks**   A variety of benchmarks have been developed to support machine learning on protein sequence and structure data. Early efforts such as TAPE (Rao et al., 2019) and ProteinNet (AlQuraishi, 2019) focused on sequence-level tasks including secondary structure prediction, contact prediction, and remote homology classification. More recently, benchmarks like PEER (Xu et al., 2022), PETA (Tan et al., 2024b), VenusFactory (Tan et al., 2025a), and ProteinGLUE (Capel et al., 2022) introduced multi-task evaluations for protein sequence understanding, emphasizing sequence-level predictions across diverse annotation types (Almagro Armenteros et al., 2017; Khurana et al., 2018). Envision (Gray et al., 2018), DeepSequence (Riesselman et al., 2018), and ProteinGym (Notin et al., 2024) advanced large-scale evaluation for fitness prediction under zero-shot or supervised mode, while FLIP (Dallago et al., 2021) curated different split strategies (e.g., one-vs-rest, which trains models on single mutations and tests on the rest of the high-order mutations) to cover various scenarios. ProteinShake (Kucera et al., 2023) standardized structural datasets and task formulations across graph, point cloud, and voxel-based representations for protein structures. ProteinBench (YE et al., 2025) evaluates various methods for different tasks such as protein structure prediction, sequence design, structure design, sequence-structure design, and molecular dynamics.

**Protein-pair Tasks**   Protein-pair modeling tasks encompass a broad spectrum of physical and functional interactions. PEER (Xu et al., 2022) includes interaction classification in human (Pan et al., 2010) and yeast (Guo et al., 2008) PPI networks, as well as affinity regression using SKEMPI (Jankauskaitė et al., 2019). Structural datasets such as MaSIF (Gainza et al., 2020) and DIPS-plus (Morehead et al., 2023) provide high-quality annotations of protein–protein interfaces, enabling geometric modeling of interaction surfaces. The recent HDPL pocketome (Moine-Franel et al., 2024) expands this scope by offering pocket-centric structural data related to PPIs and PPI-related ligand binding sites. Functional networks like STRING (Szklarczyk et al., 2019) support large-scale classification of biological associations. In the protein–ligand domain, PDBbind (Wang et al., 2005) and the Ligand Binding Affinity (LBA) tasks in Atom3D (Townshend et al., 2020) provide affinity labels derived from co-crystal structures. While these resources facilitate pairwise prediction and interaction modeling, they generally lack residue-level supervision, limiting their utility in evaluating fine-grained functional inference.

## 6   DISCUSSION AND CONCLUSION

This work introduces VENUSX, a large-scale benchmark for assessing protein representation learning at fine-grained residue and fragment resolutions. By targeting precise localized signals rather than coarse global labels, VENUSX enables rigorous evaluation of mechanistic understanding.

Our extended experiments demonstrate that traditional alignment methods (BLAST) cannot effectively transfer fine-grained functional labels (AUPR $\approx 0.04$, Appendix E), validating the necessity of deep representation learning. We further show that sequence-structure models significantly outperform sequence-only baselines in low-identity settings (e.g., MP30), suggesting that structural priors are critical for generalization when sequence homology is weak (Appendix D.4). However, current models perform poorly on epitope prediction, revealing limitations in reasoning about conformational and antibody-independent features. Cross-task correlations confirm that these fine-grained capabilities are distinct (Appendix C).

Reliability was verified through stability analysis across random seeds and ablation studies on predicted structure fidelity (Appendix D.1). VENUSX provides the community with a biologically grounded benchmark emphasizing rigorous out-of-distribution evaluation, serving as a resource for advancing functional protein modeling.

## REPRODUCIBILITY AND ETHICS STATEMENT

**Reproducibility Statement**   To ensure reproducibility, all code, data, and a leaderboard are made publicly available as open-source resources[4]. We detail our data curation in Section 2, task definitions

---

[4]Code (VenusX Github), dataset (VenusX Huggingface) and leaderboard (VenusX Website).

and splitting protocols in Section 3, and experimental setup with hyperparameters in Section 4.1 and the Appendix (Section B).

**Ethics Statement**    Our benchmark is constructed exclusively from public, anonymized protein databases and involves no human subjects. We adhere to the Code of Ethics and acknowledge that our data may reflect historical biases from these sources, a point further discussed in our dataset analysis.

**The Use of Large Language Models (LLM)**    In the preparation of this manuscript, we utilized LLMs, specifically Gemini 2.5 Pro and ChatGPT 4o, as writing assistants in the preparation of this manuscript. To refine the text, we prompted the models with role-playing instructions, such as "Act as a professional scientific editor...". The use of these tools was strictly limited to improving grammar, clarity, and overall readability. All scientific ideas, experimental results, and conclusions were conceived and formulated exclusively by the authors. All text polished or modified by the LLM was subsequently reviewed and edited by the authors to ensure that the original scientific meaning was accurately preserved.

## ACKNOWLEDGEMENTS

This work was supported by the grants from the AI for Science Program by Shanghai Municipal Commission of Economy and Informatization (2025-GZL-RGZN-BTBX-02009), National Science Foundation of China (62302291), and the National Key Research and Development Program of China (2024YFA0917603).

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

# A  DATASET

## A.1  DEFINITIONS OF INTERPRO ANNOTATION TYPES

To ensure clarity for the benchmark tasks, we provide the definitions for the functional site annotation types sourced from the InterPro database:

- **Domain**: Domains are distinct functional, structural or sequence units that may exist in a variety of biological contexts. A match to an InterPro entry of this type indicates the presence of a domain. Common examples of protein domains are the PH domain, Immunoglobulin domain or the classical C2H2 zinc finger.

- **Motif**: A short, conserved sequence of amino acids that is associated with a specific function. Motifs are typically shorter than domains and may not fold into a stable structure on their own. They often serve as signatures for a particular protein family or function.

- **Active Site**: A short sequence that contains one or more conserved residues, which allow the protein to bind to a ligand and carry out a catalytic activity.

- **Binding Site**: A short sequence that contains one or more conserved residues, which form a protein interaction site.

- **Conserved site**: A short sequence that contains one or more conserved residues, typically implying structural or functional importance.

## A.2  SEQUENCE LENGTH DISTRIBUTION

Table 7: Summary of fragment and full-sequence length statistics for each InterPro category.

| Target | Type | #Samples | Min | Max | Mean | Median |
|--------|------|----------|-----|-----|------|--------|
| Act | Fragment | 9,767 | 6 | 39 | 16.54 | 14.00 |
| | Protein | 9,667 | 39 | 2,753 | 482.51 | 379.00 |
| BindI | Fragment | 10,562 | 4 | 100 | 20.75 | 19.00 |
| | Protein | 8,959 | 45 | 2,631 | 486.54 | 415.00 |
| Evo | Fragment | 66,916 | 5 | 149 | 20.93 | 19.00 |
| | Protein | 59,948 | 16 | 2,896 | 365.88 | 305.00 |
| Dom | Fragment | 653,259 | 2 | 600 | 150.54 | 124.00 |
| | Protein | 595,443 | 16 | 2,988 | 537.41 | 444.00 |
| Motif | Fragment | 13,137 | 5 | 228 | 57.02 | 62.00 |
| | Protein | 10,271 | 30 | 2,699 | 595.15 | 558.00 |

Figures 2 to 4 and Table 7 show the length distributions of annotated protein fragments (top) and their corresponding full-length protein sequences (bottom) across five InterPro categories. To facilitate visualization, outliers were excluded: domain fragments longer than 600 residues, motif fragments exceeding 230 residues, and full-length proteins longer than 3000 residues.

Several trends can be observed:

- At the fragment level, **Act**, **BindI**, and **Evo** sites exhibit relatively short lengths, typically under 50 residues, with **Act** sites showing clear multimodal peaks due to specific catalytic motifs. **Dom** and **Motif** fragments show broader distributions, with domain fragments exhibiting a long tail.

- In contrast, full-length proteins follow a typical long-tailed distribution across all categories, with most proteins under 1000 residues but a small number extending beyond 2000. The distributions are highly skewed, especially in the **Dom** and **Evo** datasets, reflecting the diversity of protein sizes.

These distributions motivate the design of separate fragment- and full-sequence benchmarks, as the input length significantly impacts model performance and scalability.

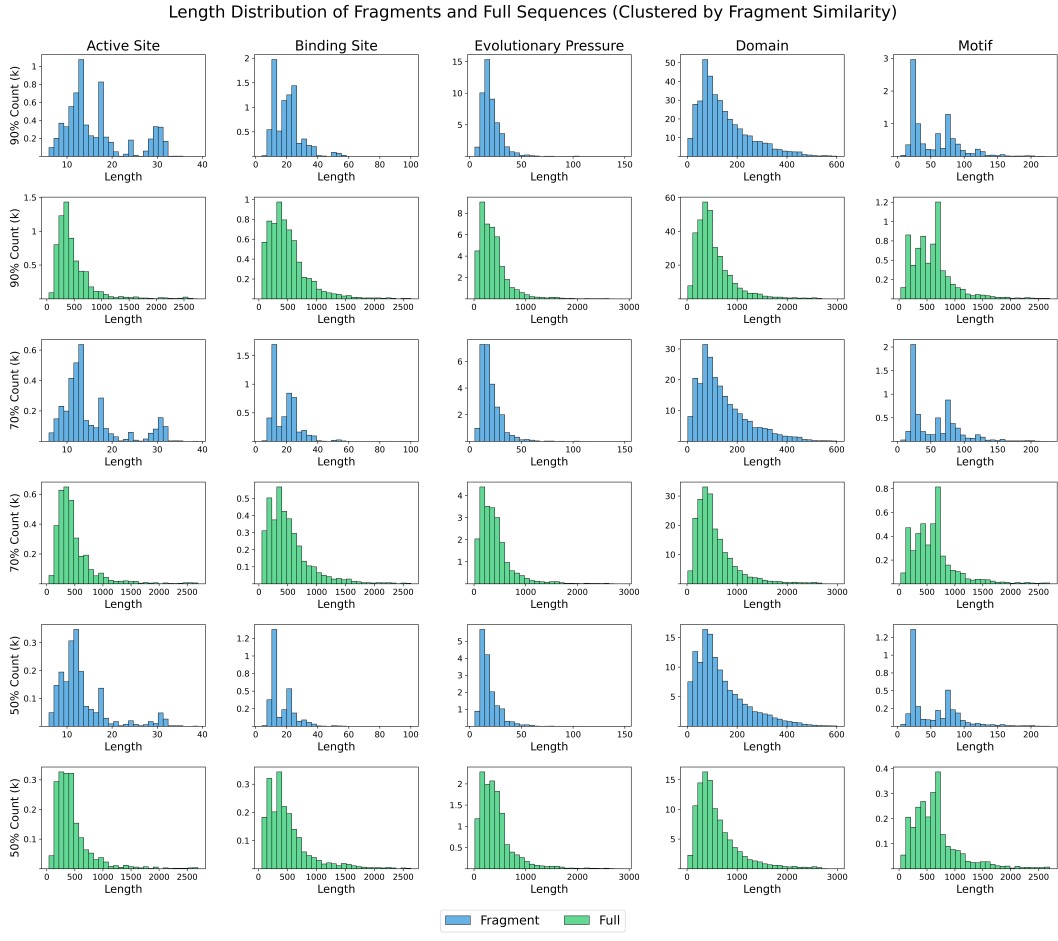

Figure 2: Sequence length distribution of the VENUSX InterPro benchmark under different levels of sequence similarity, clustered by fragment.

Table 8: Statistics of InterPro type label frequency across five categories, computed separately on annotated fragments and full sequences.

| Target | Type | #Types | Min | Max | Mean | Median |
|--------|------|--------|-----|-----|------|--------|
| **Act** | Fragment | 132 | 1 | 1176 | 73.99 | 37.50 |
| | Protein | 132 | 1 | 1172 | 73.23 | 37.50 |
| **BindI** | Fragment | 76 | 2 | 2293 | 138.97 | 53.50 |
| | Protein | 76 | 2 | 2279 | 117.88 | 53.00 |
| **Evo** | Fragment | 740 | 1 | 1418 | 90.43 | 36.00 |
| | Protein | 740 | 1 | 1074 | 81.01 | 31.50 |
| **Dom** | Fragment | 12,529 | 1 | 12002 | 52.14 | 6.00 |
| | Protein | 12,580 | 1 | 8109 | 47.33 | 6.00 |
| **Motif** | Fragment | 440 | 1 | 2222 | 29.86 | 3.00 |
| | Protein | 454 | 1 | 1247 | 22.62 | 3.00 |

## A.3 INTERPRO LABEL DISTRIBUTION

Based on Table 8, Figures 5 and 6, we analyze the distribution of residue-level labels across InterPro categories. Due to the extreme imbalance in class sizes, we apply a log-scale transformation when visualizing the number of annotations per InterPro type.

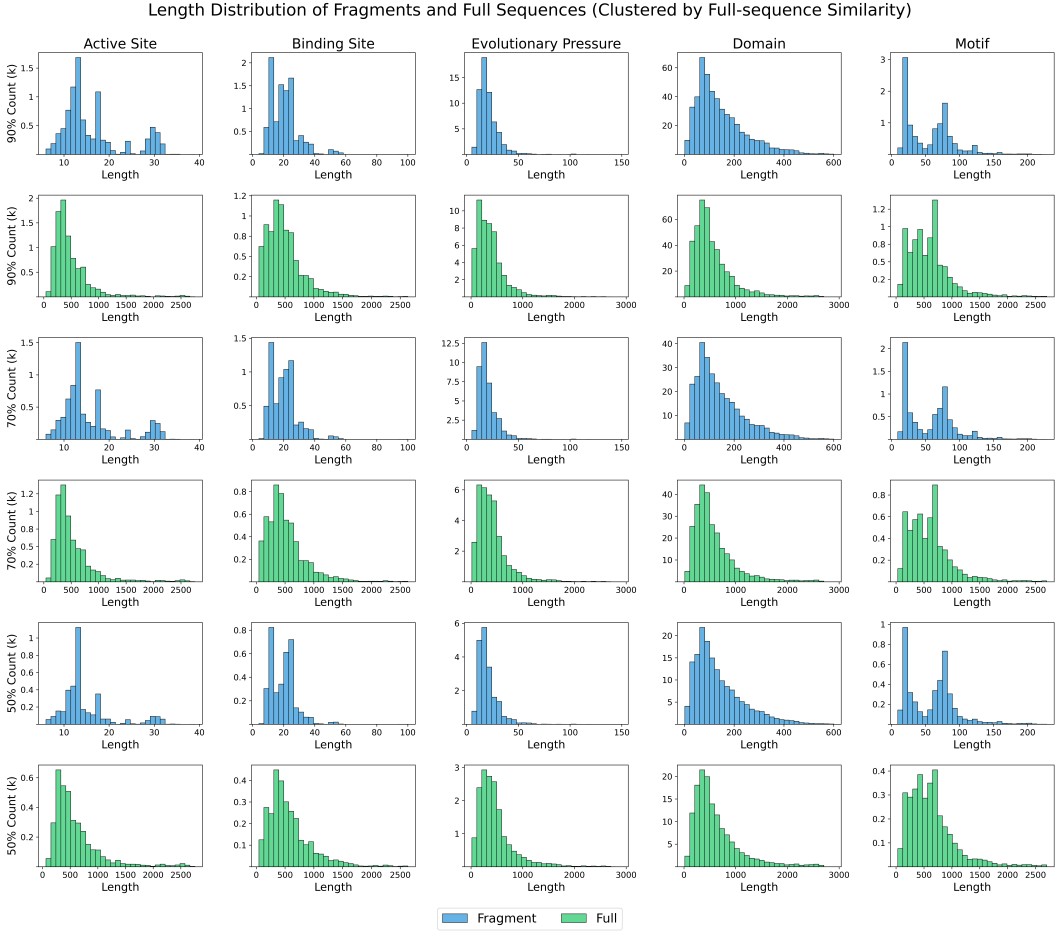

Figure 3: Sequence length distribution of the VENUSX InterPro benchmark under different levels of sequence similarity, clustered by full sequences.

- **Severe long-tail distribution across datasets.** All five datasets exhibit substantial class imbalance. For example, the domain task has over 12,000 InterPro types, but a median of only 6 annotated proteins per type at both the fragment and full-sequence level. Similarly, motif labels have a median count of 3, emphasizing the prevalence of rare classes.
- **Skewed distribution dominated by few frequent families.** In most datasets, a small number of InterPro types contribute a disproportionate number of samples. For instance, the top 5 types in the binding site fragments account for over 5,000 samples, while many other types appear fewer than 10 times.
- **Fragments exhibit slightly denser annotation than full sequences.** Across all datasets, the mean and median counts per InterPro type are consistently higher at the fragment level than for full sequences. This suggests fragments are more focused on annotated functional regions, whereas full sequences dilute sparse labels across longer chains.

These trends reinforce the importance of using macro-averaged metrics and highlight the difficulty of learning under class-imbalanced, fine-grained label regimes—especially for tasks such as domain and motif classification.

## A.4   DISTRIBUTION OF FUNCTIONAL DOMAIN START POSITIONS

Figures 7 and 8 show the distribution of relative start positions of protein functional domains. For an underlying protein, relative start position describes the normalized start location of its functional

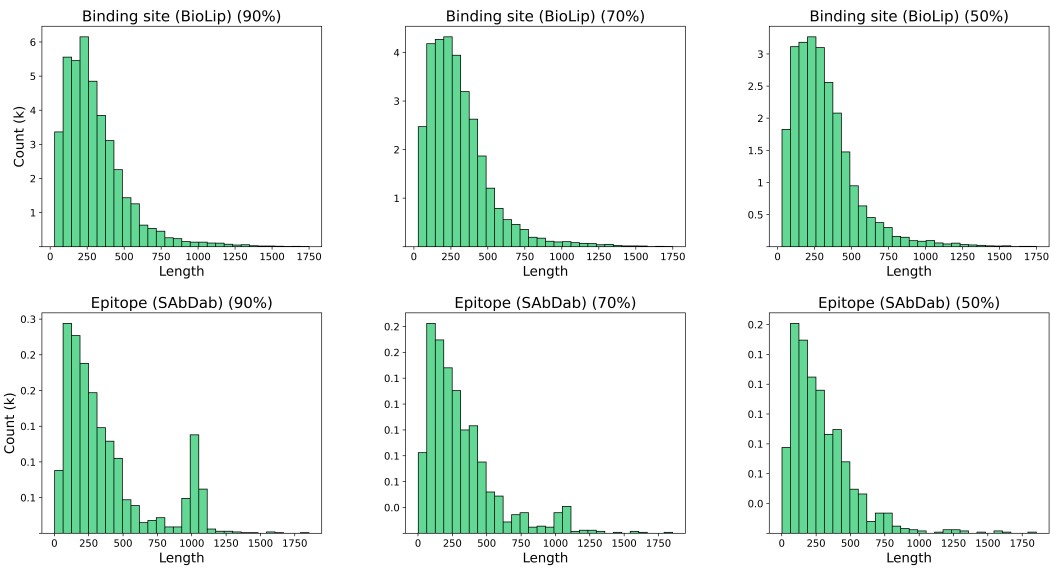

Figure 4: Sequence length distribution of BioLip and SAbDab benchmarks under different levels of sequence similarity, clustered by full sequences.

region divided by the full sequence length, with the range between 0 (start of the sequence) and 1 (end of the sequence). Distribution of starting positions reveals that different functional site types exhibit strong and distinct positional preferences along the sequence (e.g., active sites tend to appear around the middle of the sequence, while motifs favor the termini). This positional bias may introduce misleading signals for models trained on full-length sequences. When the positional distribution of functional domains in the training data is skewed toward one end of the sequence, models may overfit to these positional cues, limiting their ability to generalize to more diverse or unbiased scenarios. This underscores the need for our fine-grained benchmark to ensure that models learn true functional patterns rather than relying on location-based shortcuts.

## A.5    DATASET NUMERICAL SPLIT DETAIL

Table 9: Residue-level: Number of train/validation/test examples under mixed-family (Mix50) and cross-family splits. "Fragment" and "Protein" refer to clustering at the fragment and full-sequence levels, respectively.

| Target | Data Source | # Train/Validation/Test (Mix50) | | # Train/Validation/Test (Cross) | |
|--------|-------------|------------------|------------------|------------------|------------------|
| | | Fragment | Protein | Family | Protein |
| **Act** | InterPro (Paysan-Lafosse et al., 2023) | 1,488/186/186 | 2,929/366/367 | 104/14/14 | 7,701/880/1086 |
| **BindI** | InterPro (Paysan-Lafosse et al., 2023) | 1,640/205/205 | 2,366/296/296 | 60/8/8 | 7,729/551/679 |
| **BindB** | BioLiP (Yang et al., 2012) | – | 19,412/2,426/2,427 | – | – |
| **Evo** | InterPro (Paysan-Lafosse et al., 2023) | 10,383/1,298/1,298 | 13,552/1,694/1,694 | 592/74/74 | 48,437/5,445/6,006 |
| **Motif** | InterPro (Paysan-Lafosse et al., 2023) | 2,008/251/251 | 2,720/340/341 | 362/46/46 | 7,799/1,045/1,427 |
| **Dom** | InterPro (Paysan-Lafosse et al., 2023) | 84,489/10,561/10,562 | 113,607/14,201/14,201 | 10,065/1,258/1,260 | 477,149/56,600/61,705 |
| **Epi** | SAbDab (Dunbar et al., 2014) | – | 828/103/104 | – | – |

**Pre-filtering and Clustering.**    For all InterPro-based datasets, we apply a pre-filtering step to remove sequences lacking predicted structures from the AlphaFold Protein Structure Database (Varadi et al., 2022), ensuring structural consistency for downstream evaluations. Following this, we perform sequence identity clustering using MMseqs2 (Steinegger & Söding, 2017) under varying identity thresholds (50%, 70%, and 90%) to construct non-redundant splits at both the fragment and full-sequence levels. Clustering is conducted with a coverage mode of 1 (query coverage), and a minimum coverage of 0.8.

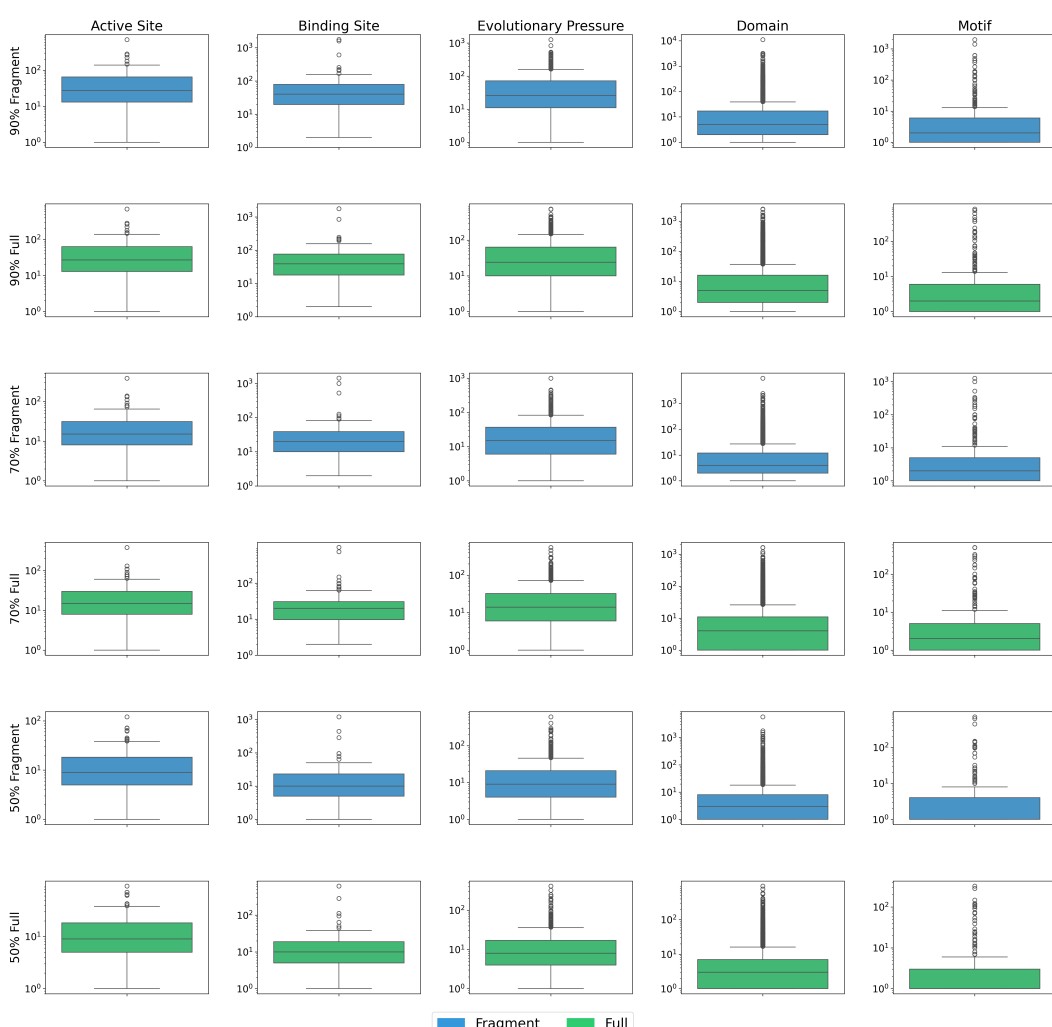

Figure 5: InterPro label distribution of the VENUSX InterPro benchmark under different levels of sequence similarity, clustered by fragment.

Table 10: Residue-level: Number of train/validation/test examples under mixed-family at 70% and 90% sequence identity. "Fragment" and "Protein" refer to clustering at the fragment and protein levels.

| Target | Data Source | # Train/Validation/Test (Mix70) | | # Train/Validation/Test (Mix90) | |
|--------|-------------|------------------|----------------|-------------------------|----------------------|
| | | Fragment | Protein | Fragment | Protein |
| Act | InterPro (Paysan-Lafosse et al., 2023) | 2,724/340/341 | 5,378/672/673 | 5,269/659/659 | 7,279/910/910 |
| BindI | InterPro (Paysan-Lafosse et al., 2023) | 3,016/377/377 | 4,432/554/555 | 5,306/663/664 | 6,428/803/804 |
| BindB | BioLiP (Yang et al., 2012) | – | 25,137/3,142/3,142 | – | 32,394/4,049/4,050 |
| Evo | InterPro (Paysan-Lafosse et al., 2023) | 17,459/2,182/2,183 | 27,848/3,481/3,481 | 33,636/4,205/4,205 | 42,869/5,359/5,359 |
| Motif | InterPro (Paysan-Lafosse et al., 2023) | 3,539/442/443 | 4,624/578/579 | 5,472/684/685 | 6,715/839/840 |
| Dom | InterPro (Paysan-Lafosse et al., 2023) | 160,201/20,025/20,026 | 215,652/26,957/26,957 | 263,870/32,984/32,984 | 348,944/43,618/43,618 |
| Epi | SAbDab (Dunbar et al., 2014) | – | 996/124/125 | – | 1,524/190/191 |

**Residue-level Split Settings.** Tables 9 and 10 summarize the number of train/validation/test examples under different split strategies. Table 9 reports counts under the mixed-family (Mix50) and cross-family splits, where fragment-level and full-sequence clustering are applied separately. Table 10 further breaks down the mixed-family splits at 70% and 90% sequence identity thresholds.

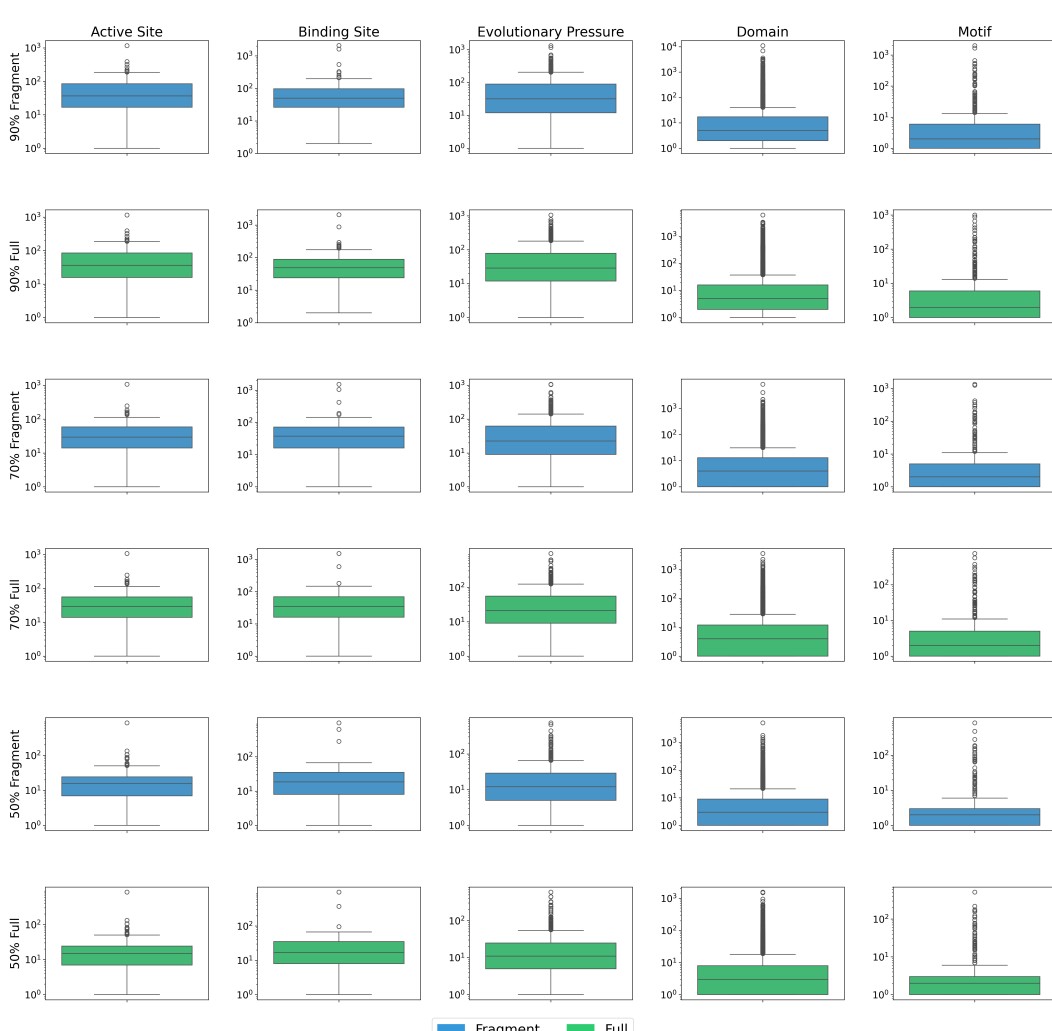

Figure 6: InterPro label distribution of the VENUSX InterPro benchmark under different levels of sequence similarity, clustered by full sequences.

Table 11: Fragment-level: Number of train/validation/test examples under mixed-family at 50%, 70%, and 90% sequence identity clustering at the fragment level.

| Target | # Train/Validation/Test | | |
|--------|-------------------------|---|---|
|        | MF50 | MF70 | MF90 |
| **Act** | 1,545/191/193 | 2,777/352/358 | 5,344/670/670 |
| **BindI** | 2,558/294/287 | 4,075/511/472 | 6,487/830/826 |
| **Evo** | 12,880/1,596/1,613 | 21,140/2,604/2,596 | 38,736/4,910/4,843 |
| **Motif** | 3,083/362/372 | 5,123/630/589 | 7,516/949/928 |
| **Dom** | 105,110/13,112/13,011 | 188,870/23,565/23,762 | 300,661/37,539/37,742 |

InterPro-based datasets support all three types of splits, while BioLiP and SAbDab only provide full-sequence annotations and thus are limited to full-protein splits. **Dom** and conserved datasets are the largest in scale, enabling more comprehensive evaluations across clustering thresholds.

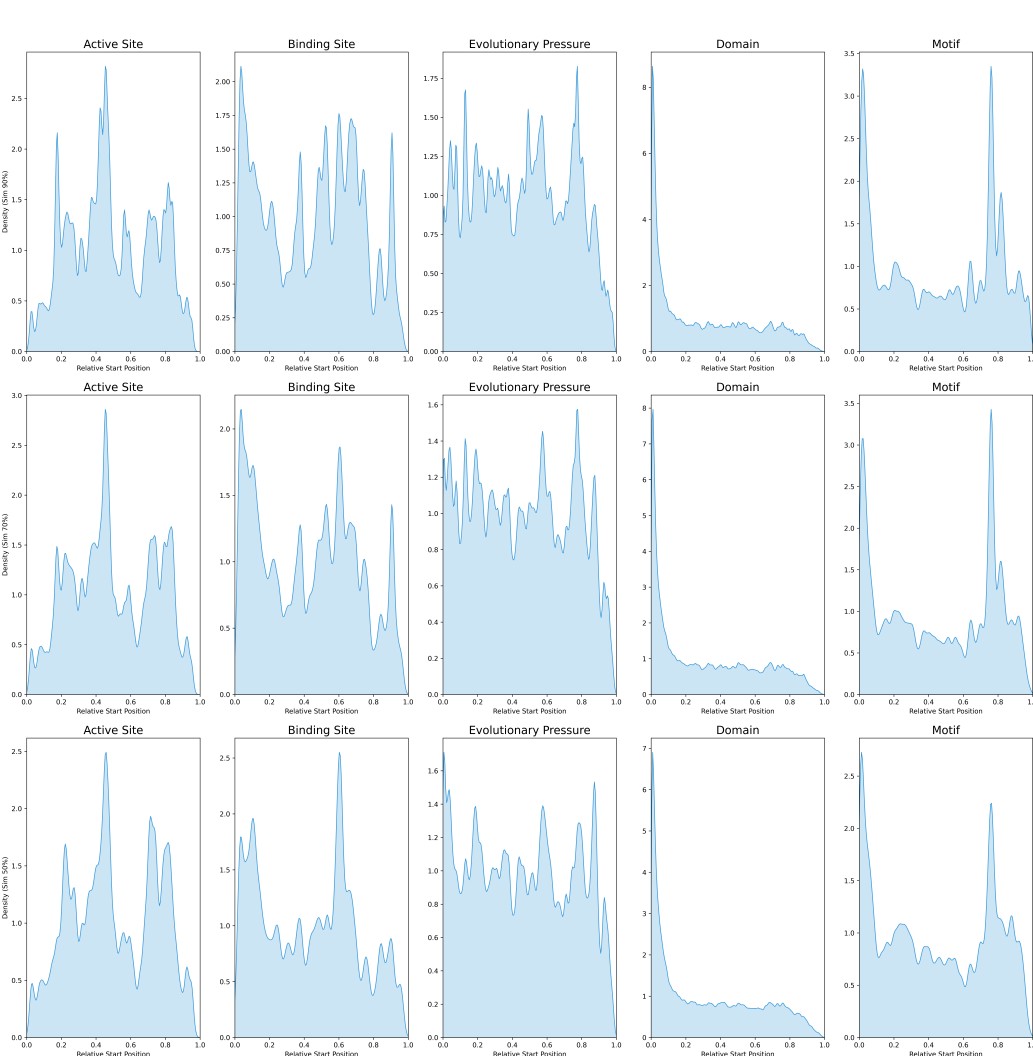

Figure 7: Protein functional domain start position density of the VENUSX InterPro benchmark under different levels of sequence similarity, clustered by fragment.

Table 12: Detailed number information of the unsupervised pair similarity evaluation task. "# Protein-P/N" and "# Frag-P/N" denote the total number of positive and negative pairs sampled by Protein sequences or fragments within InterPro families. "# Protein/Frag-pdb" denotes whether the structures of protein sequences or fragments are available.

| Target | Data Source | # Protein-P | # Protein-N | # Frag-P | # Frag-N | Protein-pdb | Frag-pdb |
|---|---|---|---|---|---|---|---|
| Act | InterPro (Paysan-Lafosse et al., 2023) | 1,314,757 | 45,405,854 | 1,331,749 | 46,360,512 | ✓ | ✓ |
| BindI | InterPro (Paysan-Lafosse et al., 2023) | 3,550,288 | 36,577,073 | 4,957,073 | 50,815,568 | ✓ | ✓ |
| Evo | InterPro (Paysan-Lafosse et al., 2023) | 7,710,941 | 1,789,140,437 | 9,990,111 | 2,228,851,959 | ✓ | ✓ |
| Motif | InterPro (Paysan-Lafosse et al., 2023) | 2,415,212 | 50,326,373 | 5,962,485 | 81,732,661 | ✓ | ✓ |
| Dom | InterPro (Paysan-Lafosse et al., 2023) | 217,681,629 | 177,064,753,702 | 346,047,386 | 215,260,712,060 | ✓ | ✓ |

**Fragment-level Split Settings.** Table 11 presents the number of train/validation/test examples across five InterPro targets under mixed-family splits with increasing sequence identity thresholds (50%, 70%, and 90%) applied at the fragment level. As expected, raising the identity threshold increases the number of retained fragments, approximately doubling the dataset size from MF50 to

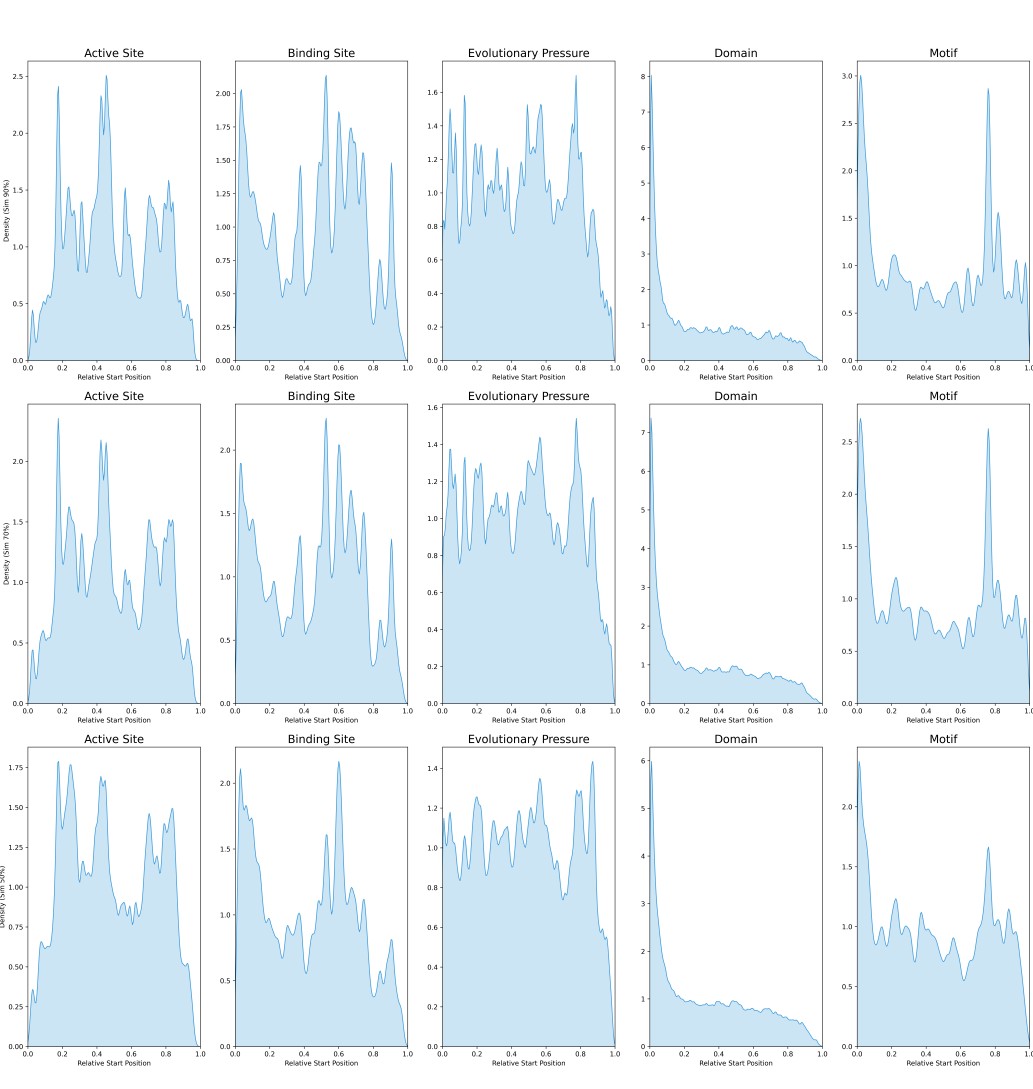

Figure 8: Protein functional domain start position density of the VENUSX InterPro benchmark under different levels of sequence similarity, clustered by full-sequence.

MF90. For instance, **Act** expands from 1.5k to 5.3k training fragments, while **Dom** scales from 105k to over 300k. This progression supports finer-grained control over redundancy and task difficulty, enabling evaluation across a spectrum of local similarity conditions. The setting facilitates analysis of model robustness to fragment diversity and homologous signal dilution.

**Pair-level Statistics.** Table 12 reports the number of positive and negative pairs for unsupervised similarity evaluation. Positive pairs share the same InterPro family, while negatives are drawn from different families. All tasks exhibit a strong imbalance, especially in large-scale domains (e.g., over 177 billion negative pairs). Structural coverage remains high across both full sequences and fragments, enabling comprehensive evaluation under both sequence- and structure-based settings.

Table 13: Summary of baseline models by input modality. "Task" indicates evaluation scope: "All" denotes all three tasks, "Sup." refers to supervised classification tasks only, and "Pair" to the pairwise functional similarity scoring task. We report model type, version, parameters, embedding size, and implementation source (via Hugging Face, GitHub, or Conda).

| Input | Model | Version | Task | # Params | # Train. Params | Embed. Dim | Implementation |
|---|---|---|---|---|---|---|---|
| Sequence-Only | ESM2 (Lin et al., 2023) | t30 | All | 150M | 410K | 640 | HF: ESM2-t30 |
| | | t33 | All | 652M | 1.6M | 1,280 | HF: ESM2-t33 |
| | | t36 | Pair. | 3,000M | – | 2,560 | HF: ESM2-t36 |
| | ESM-1B (Rives et al., 2021) | t33 | Pair. | 652M | – | 1,280 | HF: ESM-1b |
| | PROTBERT (Elnaggar et al., 2021) | uniref | All | 420M | 1.0M | 1,024 | HF: ProtBert |
| | PROTT5 (Elnaggar et al., 2021) | xl_uniref50 | Pair. | 3,000M | – | 1,024 | HF: ProtT5 |
| | ANKH (Elnaggar et al., 2023) | base | All | 450M | 591K | 768 | HF: Ankh |
| | TM-VEC (Hamamsy et al., 2024) | swiss_large | Pair. | 3,034M | – | 512 | Github: TM-vec |
| | PROSTT5 (Heinzinger et al., 2023) | AA2fold | Pair. | 3,000M | – | 1024 | HF: ProstT5 |
| | BLAST (Altschul et al., 1990) | – | Pair. | – | – | – | Conda: BLAST |
| Sequence-Structure | SAPROT (Su et al., 2023) | 35M_AF2 | All | 35M | 231K | 480 | HF: SaProt-AF2 |
| | | 650M_PDB | All | 650M | 1.6M | 1,280 | HF: SaProt-PDB |
| | PROTSSN (Tan et al., 2025c) | k20_h512 | All | 800M | 1.6M | 1,280 | HF: ProtSSN |
| | ESM-IF1 (Hsu et al., 2022) | – | Pair. | 148M | – | 512 | HF: ESM-IF1 |
| | MIF-ST (Yang et al., 2023) | – | Pair. | 643M | – | 256 | Github: MIF-ST |
| | FOLDSEEK (Van Kempen et al., 2024) | 3Di-AA | Pair. | – | – | – | Conda: Foldseek |
| Structure-Only | GVP-GNN (Jing et al., 2021) | 3-layers | Sup. | 3M | 3M | 512 | GitHub: GVP |
| | FOLDSEEK (Van Kempen et al., 2024) | 3Di | Pair. | – | – | – | Conda: Foldseek |
| | TM-ALIGN (Zhang & Skolnick, 2005) | mean | Pair. | – | – | – | Conda: TM-align |

# B  BASELINES

## B.1  DEEP LEARNING MODELS

**Sequence-Only.** Sequence-only baselines include both encoder-only and encoder–decoder architectures. Encoder-only models such as ESM2 (t30, t33, t36) (Lin et al., 2023), ESM-1B (Rives et al., 2021), and PROTBERT (Elnaggar et al., 2021) are pretrained protein language models using masked language modeling on large sequence corpora. ANKH (Elnaggar et al., 2023) and PROTT5 (Elnaggar et al., 2021), in contrast, adopt encoder–decoder architectures, enabling bidirectional contextualization and autoregressive decoding. While TM-VEC (Hamamsy et al., 2024) and PROSTT5 (Heinzinger et al., 2023) only require sequence inputs, both incorporate structural inductive signals during training: TM-VEC is trained to regress TM-scores, and PROSTT5 is fine-tuned to translate Foldseek-derived structural tokens.

**Sequence–Structure.** Sequence–structure models combine sequence and structural information in diverse ways. SAPROT (Su et al., 2023) fuses amino acid tokens with Foldseek-derived structural tokens and is trained using multi-modal masked language modeling. PROTSSN (Tan et al., 2025c) integrates ESM2 (Lin et al., 2023) embeddings with geometric graph neural networks, enabling joint sequence–structure representation learning. Both ESM-IF1 (Hsu et al., 2022) and MIF-ST (Yang et al., 2023) are inverse folding models: ESM-IF1 is pretrained on large-scale backbone recovery, while MIF-ST uses structure-conditioned geometric networks initialized from large protein transformers.

**Structure-Only.** Structure-only baselines rely purely on 3D geometric inputs. GVP-GNN (Jing et al., 2021) is a non-pretrained geometric deep learning model that uses residue type and atomic coordinate features for message passing.

## B.2  ALIGNMENT-BASED METHODS

**Foldseek.** We employ FOLDSEEK (Van Kempen et al., 2024) to evaluate structural similarity between query and target proteins under two alignment modes: *3Di-only* (`-alignment-type 0`) and *3Di+AA* (`-alignment-type 2`). To maximize sensitivity, we activate exhaustive pairwise comparison via `-exhaustive-search`, set a high `-e` threshold of 1,000, and use `-min-seq-id 0.0` to allow all sequence identity levels. We retain up to 100,000 alignments per query using `-max-seqs 100000`, and parallelize computation across available CPU threads. The output alignment scores are used to compute similarity for unsupervised pair-level evaluation (e.g., AUC). FOLDSEEK achieves state-of-the-art tradeoffs between alignment speed and accuracy for large-scale protein structure comparison.

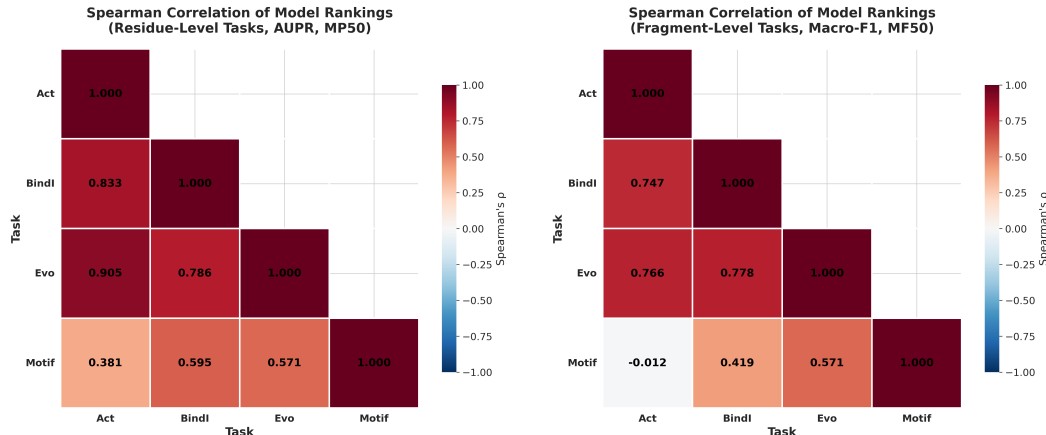

(a) Residue-Level Correlation (AUPR, MP50)    (b) Fragment-Level Correlation (Macro-F1, MF50)

Figure 9: Spearman's rank correlation ($\rho$) of model performance across tasks. **(a)** Correlation of AUPR scores for Residue-Level tasks on the MP50 split. We observe strong correlations between Act, BindI, and Evo tasks. **(b)** Correlation of Macro-F1 scores for Fragment-Level tasks on the MF50 split. We find a striking zero correlation between Act and Motif tasks.

**BLASTp.** We adopt BLAST (Altschul et al., 1990) as a classical sequence-based alignment baseline. Our setup disables low-complexity masking (`-seg no`) to preserve short functional regions and sets a permissive $E$-value threshold (`-evalue 1000000`) to retain weak similarities. Word size is reduced to 2 (`-word_size 2`) to improve alignment sensitivity, and a large hit buffer (`-max_target_seqs 100000000`) ensures comprehensive coverage. The output is recorded in tabular format (`-outfmt 6`), including sequence IDs, identity, alignment length, mismatches, gaps, and bit scores. BLAST is used as a baseline for fragment-level and full-sequence pair similarity evaluation.

**TM-align.** To benchmark structure-only alignment, we apply TM-ALIGN (Zhang & Skolnick, 2005) on PDB-format query and reference proteins via the default command-line interface (`TMalign query.pdb ref.pdb`). TM-ALIGN returns key statistics, including RMSD, sequence identity, aligned length, and two TM-scores (normalized by query and reference, respectively). We record the average TM-score between the two directions as the final similarity metric for evaluation. TM-ALIGN is widely regarded as a reliable tool for structure-based homology assessment, though its quadratic computational complexity limits scalability on large benchmarks.

## C  CROSS-TASK CORRELATION ANALYSIS

We performed a cross-task correlation analysis to determine if model strengths are task-specific or general. We computed the Spearman's rank correlation ($\rho$) on the model performance scores (AUPR for residue tasks [MP50 split], Macro-F1 for fragment tasks [MF50 split]). The results are visualized in the heatmaps in Figure 9.

Residue-Level Analysis (Figure 9a): We observe strong, positive correlations across the Act (Active Site), BindI (Binding Site), and Evo (Evolutionary Pressure) tasks ($\rho$ ranging from 0.79 to 0.90). This is intuitive and confirms the validity of our tasks, as active and binding sites are, by definition, under high evolutionary pressure and functionally related. Models that excel at identifying one typically excel at the others. The Motif task shows a weaker correlation, suggesting that identifying these (often longer) fragments is a partially distinct capability.

Fragment-Level Analysis (Figure 9b): A similar pattern emerges for fragment classification, where Act, BindI, and Evo tasks are highly correlated ($\rho \approx 0.75 - 0.78$). However, we find a striking result: there is effectively zero correlation ($\rho = -0.01$) between the Act and Motif tasks. This empirically demonstrates that the model features required to classify a general "Motif" are completely different from those required to classify a specific "Active Site motif."

Table 14: Ablation study on structure quality. **(a)** Performance comparison (AUPR) of SaProt models on the Epitope (Epi) task using experimental PDB structures versus AlphaFold2 (AF2) predicted structures. Both settings use the MP50 split. **(b)** Distribution of pLDDT confidence scores for the AlphaFold structures used in VENUSX, indicating high structural reliability.

| **(a) Model Performance** | | | | **(b) AlphaFold pLDDT Statistics** | | | | | |
|---|---|---|---|---|---|---|---|---|---|
| **Model** | **PDB** | **AF2** | | **Level** | **Mean** | **Min** | **Max** | **25%** | **50%** |
| SaProt 35M | 0.1820 | 0.1721 | | Protein | 91.01 | 42.00 | 98.71 | 89.10 | 92.70 |
| SaProt 650M | 0.1948 | 0.1889 | | Residue | 91.40 | 19.44 | 99.00 | 89.75 | 94.88 |

Table 15: AUPR comparison between GVP-GNN (from scratch) and GearNet (pre-trained) on the Epitope task.

| **Model** | **Res_Epi_MP50** | **Res_Epi_MP70** | **Res_Epi_MP90** |
|---|---|---|---|
| GVP-GNN | 0.118 | 0.139 | 0.196 |
| GearNet | 0.116 | 0.129 | 0.167 |

# D   ABLATION STUDIES

## D.1   IMPACT OF STRUCTURE SOURCE (PREDICTED VS. EXPERIMENTAL)

To address concerns regarding the use of predicted structures (AlphaFold DB) instead of experimental ones (PDB), we performed an ablation study on the Epitope (Epi) task. We evaluated the structure-aware SaProt models on the subset of the MP50 split where experimental PDB structures were available, and compared this to the standard setting using AlphaFold structures.

**Results**: Table 14(a) presents the performance comparison. We observe that the AUPR scores are highly consistent between the two settings. For instance, SaProt-650M achieves an AUPR of 0.1948 on PDB structures and 0.1889 on AlphaFold structures. This minimal performance difference is attributable to the high confidence of the predicted structures used in VENUSX. As shown in Table 14(b), the mean pLDDT score is >91 at both the protein and residue levels, with a median of 94.88 for residues. This indicates that the predicted structures are of sufficient quality to serve as a reliable proxy for experimental structures, justifying their use to scale up the benchmark.

## D.2   EVALUATION OF PRE-TRAINED STRUCTURE-ONLY MODELS

To assess the impact of structural pre-training, we compared GearNet (pre-trained) against GVP-GNN (trained from scratch) on the Residue-level Epitope task (Res_Epi). As shown in Table 15, GearNet performs comparably to GVP-GNN across all splits (e.g., 0.116 vs. 0.118 on MP50). This indicates that for this specific task relying on fine-grained surface properties, general structural pre-training objectives do not immediately outperform supervised geometric learning.

## D.3   TRAINING STABILITY AND RANDOM SEED SENSITIVITY

To verify that our benchmark results are robust to random initialization and not artifacts of a specific seed, we analyzed the stability of model performance.

**Methodology**: We re-trained six representative baseline models (ESM2, ProtBert, ProtT5, Ankh) on two of the most challenging datasets: Res_Act_MP30 and Res_BindI_MP30. Each model was trained independently using three different random seeds: 42, 3407 (our default), and 12345.

**Results**: Table 16 summarizes the Mean Recall and Standard Deviation (Std) for each model. We observe:

- Low Variance: The standard deviations are consistently low across all models and tasks, typically ranging from 0.003 to 0.014.

Table 16: Stability analysis of model performance (Recall) across three different random seeds (42, 3407, 12345) on the challenging MP30 split. We report the Mean $\pm$ Standard Deviation. The consistently low standard deviation ($< 0.015$) across diverse models and tasks confirms that the benchmark rankings are robust to random initialization.

| | Res_Act_MP30 | | Res_BindI_MP30 | |
|---|---|---|---|---|
| **Model** | **Mean** | **Std** | **Mean** | **Std** |
| ESM2 t30 | 0.765 | 0.011 | 0.818 | 0.003 |
| ESM2 t33 | 0.785 | 0.004 | 0.826 | 0.012 |
| ProtBert | 0.587 | 0.007 | 0.668 | 0.014 |
| ProtT5 xl_uniref50 | 0.767 | 0.006 | 0.883 | 0.010 |
| Ankh base | 0.798 | 0.010 | 0.895 | 0.008 |
| Ankh large | 0.776 | 0.010 | 0.842 | 0.008 |

Table 17: Performance comparison (**Recall**) on Active Site (Act) and Evolutionary Pressure (Evo) tasks.

| Model | Res_Act_MP30 | Res_Act_MP50 | Res_Evo_MP30 | Res_Evo_MP50 |
|---|---|---|---|---|
| ESM2 t33 (Seq-only) | 0.787 | **0.870** | 0.703 | 0.861 |
| SaProt 650M (Seq-Struct) | **0.792** | 0.850 | **0.714** | **0.866** |

- Stable Rankings: The relative performance ranking of different models remains consistent across seeds. For example, Ankh base consistently outperforms ProtBert by a significant margin on both tasks, regardless of the seed.

These results confirm that the performance differences observed in VENUSX, are driven by the quality of the representations and the model architectures, rather than hyperparameter noise or initialization luck.

### D.4 ANALYSIS OF ROBUSTNESS TO LOW SEQUENCE IDENTITY

Sequence-structure hybrid models (e.g., SaProt) exhibit superior generalization on cross-family (OOD) splits compared to sequence-only models. To investigate the mechanism behind this, we analyzed model performance under decreasing levels of sequence identity, isolating the effect of "low homology" which characterizes remote generalization.

**Hypothesis:** We hypothesized that as sequence identity drops, the predictive signal from sequence alone weakens. However, since structural folds are often conserved even when sequences diverge, structure-aware models should be more robust in low-identity regimes.

**Results:** We evaluated models on two mix-family splits with different difficulty levels: **MP50** (50% identity) and the significantly harder **MP30** (30% identity). Table 17 summarizes the **Recall** scores.

- **Moderate Homology (MP50):** On the MP50 split, the sequence-only ESM2 model performs comparably to or slightly better than SaProt (e.g., Recall of 0.870 vs. 0.850 on Act). This suggests that when sequence homology is sufficient, sequence features alone are highly effective.

- **Low Homology (MP30):** When the sequence identity threshold is tightened to 30%, the advantage shifts. SaProt consistently outperforms ESM2 on both tasks, achieving a higher Recall of **0.792 vs. 0.787** on Act, and **0.714 vs. 0.703** on Evo.

**Conclusion:** These findings clarify why sequence-structure models excel in cross-family tasks. While sequence features dominate in high-homology settings, structural information provides a critical signal when sequence similarity fades, enabling better generalization to remote homologs.

Table 18: Performance of the BLAST baseline on residue-level active site classification (Act). We report the specific AUPR scores for the top-3 runs out of 10 independent trials. Even the best-case results (~0.05) are drastically lower than deep learning models (>0.85), confirming that alignment is insufficient for this task.

| P-Identity Threshold | Res_Act_MF50 | | | Res_Act_MF70 | | |
|---|---|---|---|---|---|---|
| | Run 1 | Run 2 | Run 3 | Run 1 | Run 2 | Run 3 |
| 100 (Exact Match) | 0.0407 | 0.0410 | 0.0417 | 0.0471 | 0.0480 | 0.0492 |
| 90 (High Identity) | 0.0435 | 0.0431 | 0.0439 | 0.0497 | 0.0528 | 0.0520 |

# E    NON-DEEP LEARNING BASELINE PERFORMANCE

To assess whether fine-grained functional residues can be accurately identified solely through sequence homology transfer, we evaluated a traditional alignment-based baseline for the residue-level active site classification task (Act). This baseline utilizes NCBI BLAST+ to transfer functional annotations from the training set to the test set.

## E.1    METHODOLOGY

- **Query Generation:** We implemented a stratified sampling strategy to construct the query database. For each experimental run, randomly select 100 distinct InterPro IDs from the training set. For each selected ID, we randomly sampled 100 sequences (or all available sequences if the family size was smaller). From these sampled sequences, we extracted all continuous sub-sequences annotated as positive (label=1) to serve as functional motif queries.

- **Search Strategy:** These fragments were compiled into a query database and searched against the full-length protein sequences in the test set. We utilized the `blastp` algorithm with the `task=blastp-short` option, which is optimized for aligning short sequences that might otherwise be missed by standard parameters due to low complexity or short length.

- **Scoring Mechanism:** For every residue in a test sequence covered by a significant BLAST hit, we assigned a prediction score equal to the percentage identity (`pident` / 100.0) of that alignment. If a residue was covered by multiple overlapping hits, the maximum score was retained to represent the highest confidence match. Residues not covered by any alignment were assigned a score of 0.

- **Robust Evaluation:** To minimize sampling variance and estimate an upper bound for alignment-based performance, we independently repeated this procedure 10 times using different random subsets of queries (different IDs and sequences). We report the specific AUPR scores of the **top-3 performing runs**.

## E.2    RESULTS AND ANALYSIS

Table 18 summarizes the performance on the mixed-family splits (MF50 and MF70) under strict (`pident=100`) and relaxed (`pident=90`) filtering thresholds.

As shown in Table 18, the alignment-based method yields extremely low AUPR scores. Even in the best-performing runs, the AUPR remains approximately 0.04–0.05 (e.g., 0.0528 for the best run on Act_MF70). This poor performance stands in stark contrast to deep learning models (e.g., ESM2, SaProt), which consistently achieve AUPR scores exceeding 0.85 on the same tasks (see Table 4).

This significant performance disparity highlights the limitations of simple homology-based label transfer for fine-grained residue-level prediction. While BLAST is effective for global homology detection, it struggles to precisely map functional residues when the local sequence motif exhibits variability or degenerate patterns, even within homologous families. These results empirically validate the necessity of VENUSX, and the representation learning approach, which can capture complex, non-linear functional signals that alignment methods miss.

Table 19: Detailed residue-level classification performance across **BindB** and **Epi** datasets and data splits. "MP50", "MP70", and "MP90" refer to mixed-family splits with 50%, 70%, and 90% sequence identity filtering applied at the full-sequence level. Metrics reported include AUPR, Precision, Recall, F1 scores for negative and positive classes, and Macro-F1.

| Metric | Model | BindB | | | Epi | | |
| --- | --- | --- | --- | --- | --- | --- | --- |
| | | MP50 | MP70 | MP90 | MP50 | MP70 | MP90 |
| AUPR | ESM2 t30 | 0.408 | 0.465 | 0.496 | 0.186 | 0.184 | 0.277 |
| | ESM2 t33 | 0.446 | 0.494 | 0.535 | 0.174 | 0.200 | 0.290 |
| | ProtBert | 0.340 | 0.410 | 0.466 | 0.169 | 0.177 | 0.266 |
| | Ankh | 0.421 | 0.487 | 0.527 | 0.167 | 0.215 | 0.270 |
| precision | ESM2 t30 | 0.598 | 0.637 | 0.674 | 1.0 | 0.384 | 0.545 |
| | ESM2 t33 | 0.605 | 0.646 | 0.675 | 0.0 | 1.0 | 0.512 |
| | ProtBert | 0.547 | 0.619 | 0.706 | 1.0 | 0.432 | 0.534 |
| | Ankh | 0.634 | 0.660 | 0.677 | 0.0 | 1.0 | 0.571 |
| Recall | ESM2 t30 | 0.289 | 0.317 | 0.316 | 0.001 | 0.043 | 0.091 |
| | ESM2 t33 | 0.329 | 0.356 | 0.386 | 0.0 | 0.003 | 0.139 |
| | ProtBert | 0.238 | 0.264 | 0.257 | 0.001 | 0.005 | 0.072 |
| | Ankh | 0.260 | 0.335 | 0.357 | 0.0 | 0.008 | 0.054 |
| F1-Negative | ESM2 t30 | 0.987 | 0.986 | 0.986 | 0.958 | 0.960 | 0.968 |
| | ESM2 t33 | 0.987 | 0.987 | 0.987 | 0.958 | 0.961 | 0.968 |
| | ProtBert | 0.986 | 0.986 | 0.986 | 0.958 | 0.961 | 0.968 |
| | Ankh | 0.987 | 0.987 | 0.987 | 0.958 | 0.961 | 0.968 |
| F1-Positive | ESM2 t30 | 0.390 | 0.423 | 0.430 | 0.002 | 0.077 | 0.156 |
| | ESM2 t33 | 0.427 | 0.459 | 0.491 | 0.0 | 0.006 | 0.218 |
| | ProtBert | 0.332 | 0.370 | 0.377 | 0.002 | 0.010 | 0.126 |
| | Ankh | 0.369 | 0.444 | 0.483 | 0.0 | 0.002 | 0.098 |
| Macro-F1 | ESM2 t30 | 0.689 | 0.705 | 0.708 | 0.480 | 0.518 | 0.562 |
| | ESM2 t33 | 0.707 | 0.723 | 0.739 | 0.479 | 0.484 | 0.593 |
| | ProtBert | 0.659 | 0.678 | 0.682 | 0.480 | 0.486 | 0.547 |
| | Ankh | 0.678 | 0.715 | 0.735 | 0.479 | 0.489 | 0.533 |

## F  DETAILED EXPERIMENTAL RESULTS

Here, we provide detailed experimental results for two tasks: Residue-Level Binary Classification and Fragment-Level Multi-Class Classification. We report all evaluation metrics recorded during the experiments to offer a comprehensive assessment of model performance across different aspects. Please refer to Tables 19–22 for the complete results.

Model scores on the mix-family tasks are relatively high. To further demonstrate the benchmark's remaining headroom, Table 23 presents new results from a more challenging experiment conducted under a stricter "MP30" split (30% sequence identity). The results reveal substantial performance variance across different models, and overall performance generally declines, with the best model achieving an average Recall of 0.73 across the five test sets.

To evaluate the impact of data on model performance, we analyzed how model accuracy varies with the sequence similarity of the training data in **Act** and **Evo** tasks. For each task, we filtered the datasets using sequence identity thresholds of 90%, 70%, 50%, and 30%. Subsets with lower sequence identity thresholds contain fewer training and testing samples (see the statistics in Tables 9 and 10). As shown in Table 24, model generalization performance decreases progressively as both the sequence similarity threshold and the dataset size are reduced.

Table 20: Detailed residue-level classification performance across **Act**, **BindI**, and **Evo** datasets and data splits. "MF50" and "MP50" refer to mixed-family splits with 50% sequence identity filtering applied at the fragment and full-sequence levels, respectively. Metrics reported include AUPR, Precision, Recall, F1 scores for negative and positive classes, and Macro-F1.

| Metric | Model | Act | | | BindI | | | Evo | | |
|---|---|---|---|---|---|---|---|---|---|---|
| | | MF50 | MP50 | Cross | MF50 | MP50 | Cross | MF50 | MP50 | Cross |
| AUPR | ESM2 t30 | 0.855 | 0.932 | 0.143 | 0.912 | 0.963 | 0.133 | 0.862 | 0.897 | 0.235 |
| | ESM2 t33 | 0.852 | 0.955 | 0.143 | 0.904 | 0.971 | 0.159 | 0.899 | 0.926 | 0.262 |
| | ProtBert | 0.764 | 0.895 | 0.131 | 0.857 | 0.926 | 0.112 | 0.771 | 0.803 | 0.243 |
| | Ankh base | 0.873 | 0.895 | 0.166 | 0.907 | 0.970 | 0.145 | 0.895 | 0.932 | 0.275 |
| | GVP-GNN | 0.523 | 0.896 | 0.101 | 0.611 | 0.883 | 0.040 | 0.342 | 0.792 | 0.101 |
| | SaProt 35M | 0.688 | 0.905 | 0.114 | 0.807 | 0.927 | 0.230 | 0.724 | 0.772 | 0.272 |
| | SaProt 650M | 0.745 | 0.945 | 0.185 | 0.838 | 0.960 | 0.182 | 0.734 | 0.912 | 0.274 |
| | ProtSSN | 0.465 | 0.917 | 0.156 | 0.801 | 0.907 | 0.095 | 0.715 | 0.895 | 0.227 |
| Precision | ESM2 t30 | 0.826 | 0.851 | 0.278 | 0.859 | 0.915 | 0.525 | 0.816 | 0.879 | 0.374 |
| | ESM2 t33 | 0.845 | 0.851 | 0.126 | 0.869 | 0.905 | 0.581 | 0.856 | 0.856 | 0.403 |
| | ProtBert | 0.791 | 0.833 | 0.131 | 0.855 | 0.897 | 0.416 | 0.805 | 0.803 | 0.482 |
| | Ankh base | 0.862 | 0.873 | 0.190 | 0.849 | 0.919 | 0.437 | 0.882 | 0.932 | 0.387 |
| | GVP-GNN | 0.735 | 0.824 | 0.019 | 0.730 | 0.874 | 0.0 | 0.810 | 0.781 | 0.176 |
| | SaProt 35M | 0.818 | 0.879 | 0.132 | 0.813 | 0.902 | 0.634 | 0.819 | 0.841 | 0.382 |
| | SaProt 650M | 0.812 | 0.845 | 0.241 | 0.827 | 0.900 | 0.661 | 0.809 | 0.828 | 0.456 |
| | ProtSSN | 0.523 | 0.835 | 0.241 | 0.818 | 0.887 | 0.379 | 0.790 | 0.815 | 0.452 |
| Recall | ESM2 t30 | 0.676 | 0.793 | 0.060 | 0.859 | 0.897 | 0.078 | 0.783 | 0.750 | 0.097 |
| | ESM2 t33 | 0.682 | 0.848 | 0.031 | 0.830 | 0.924 | 0.108 | 0.806 | 0.755 | 0.122 |
| | ProtBert | 0.565 | 0.750 | 0.020 | 0.694 | 0.839 | 0.048 | 0.610 | 0.597 | 0.009 |
| | Ankh base | 0.700 | 0.864 | 0.025 | 0.866 | 0.922 | 0.086 | 0.735 | 0.744 | 0.169 |
| | GVP-GNN | 0.362 | 0.798 | 0.001 | 0.519 | 0.788 | 0.0 | 0.091 | 0.718 | 0.035 |
| | SaProt 35M | 0.408 | 0.733 | 0.036 | 0.705 | 0.822 | 0.135 | 0.520 | 0.649 | 0.172 |
| | SaProt 650M | 0.511 | 0.850 | 0.072 | 0.768 | 0.918 | 0.135 | 0.554 | 0.700 | 0.111 |
| | ProtSSN | 0.209 | 0.801 | 0.014 | 0.705 | 0.788 | 0.029 | 0.507 | 0.852 | 0.034 |
| F1-Negative | ESM2 t30 | 0.992 | 0.995 | 0.967 | 0.993 | 0.996 | 0.975 | 0.988 | 0.990 | 0.957 |
| | ESM2 t33 | 0.993 | 0.996 | 0.964 | 0.992 | 0.996 | 0.976 | 0.990 | 0.992 | 0.957 |
| | ProtBert | 0.991 | 0.994 | 0.967 | 0.989 | 0.994 | 0.974 | 0.984 | 0.986 | 0.960 |
| | Ankh base | 0.993 | 0.996 | 0.968 | 0.992 | 0.996 | 0.974 | 0.989 | 0.992 | 0.955 |
| | GVP-GNN | 0.989 | 0.995 | 0.969 | 0.982 | 0.993 | 0.975 | 0.973 | 0.986 | 0.955 |
| | SaProt 35M | 0.990 | 0.995 | 0.964 | 0.988 | 0.994 | 0.976 | 0.982 | 0.985 | 0.955 |
| | SaProt 650M | 0.986 | 0.996 | 0.965 | 0.989 | 0.996 | 0.977 | 0.983 | 0.991 | 0.959 |
| | ProtSSN | 0.988 | 0.995 | 0.969 | 0.988 | 0.993 | 0.975 | 0.982 | 0.991 | 0.960 |
| F1-Positive | ESM2 t30 | 0.744 | 0.821 | 0.098 | 0.859 | 0.906 | 0.136 | 0.799 | 0.810 | 0.154 |
| | ESM2 t33 | 0.755 | 0.850 | 0.050 | 0.849 | 0.915 | 0.181 | 0.831 | 0.858 | 0.187 |
| | ProtBert | 0.659 | 0.789 | 0.035 | 0.766 | 0.867 | 0.086 | 0.694 | 0.701 | 0.017 |
| | Ankh base | 0.773 | 0.869 | 0.045 | 0.857 | 0.920 | 0.144 | 0.802 | 0.858 | 0.235 |
| | GVP-GNN | 0.485 | 0.810 | 0.002 | 0.607 | 0.829 | 0.0 | 0.164 | 0.748 | 0.058 |
| | SaProt 35M | 0.544 | 0.800 | 0.056 | 0.755 | 0.860 | 0.223 | 0.636 | 0.689 | 0.238 |
| | SaProt 650M | 0.627 | 0.848 | 0.110 | 0.796 | 0.909 | 0.224 | 0.658 | 0.851 | 0.178 |
| | ProtSSN | 0.329 | 0.818 | 0.026 | 0.757 | 0.839 | 0.053 | 0.618 | 0.833 | 0.062 |
| Macro-F1 | ESM2 t30 | 0.868 | 0.908 | 0.533 | 0.926 | 0.951 | 0.556 | 0.894 | 0.900 | 0.555 |
| | ESM2 t33 | 0.874 | 0.923 | 0.507 | 0.921 | 0.955 | 0.579 | 0.910 | 0.925 | 0.572 |
| | ProtBert | 0.825 | 0.892 | 0.501 | 0.878 | 0.931 | 0.530 | 0.839 | 0.843 | 0.489 |
| | Ankh base | 0.883 | 0.933 | 0.507 | 0.925 | 0.958 | 0.559 | 0.896 | 0.925 | 0.595 |
| | GVP-GNN | 0.736 | 0.903 | 0.485 | 0.795 | 0.911 | 0.488 | 0.569 | 0.867 | 0.506 |
| | SaProt 35M | 0.767 | 0.897 | 0.510 | 0.871 | 0.927 | 0.599 | 0.809 | 0.837 | 0.596 |
| | SaProt 650M | 0.808 | 0.922 | 0.538 | 0.893 | 0.953 | 0.600 | 0.820 | 0.921 | 0.568 |
| | ProtSSN | 0.658 | 0.906 | 0.498 | 0.873 | 0.911 | 0.514 | 0.800 | 0.912 | 0.511 |

Table 21: Detailed residue-level classification performance across **Motif** and **Dom** datasets and data splits. "MF50" and "MP50" refer to mixed-family splits with 50% sequence identity filtering applied at the fragment and full-sequence levels, respectively. Metrics reported include AUPR, Precision, Recall, F1 scores for negative and positive classes, and Macro-F1.

| Metric | Model | Motif | | | Dom | | |
|---|---|---|---|---|---|---|---|
| | | MF50 | MP50 | Cross | MF50 | MP50 | Cross |
| AUPR | ESM2 t30 | 0.855 | 0.850 | 0.433 | 0.634 | 0.634 | 0.470 |
| | ESM2 t33 | 0.874 | 0.857 | 0.456 | 0.666 | 0.657 | 0.506 |
| | ProtBert | 0.779 | 0.796 | 0.348 | 0.591 | 0.592 | 0.508 |
| | Ankh base | 0.884 | 0.870 | 0.394 | 0.673 | 0.665 | 0.449 |
| | GVP-GNN | 0.661 | 0.736 | 0.329 | 0.560 | 0.557 | 0.468 |
| | SaProt 35M | 0.767 | 0.784 | 0.408 | 0.574 | 0.584 | 0.525 |
| | SaProt 650M | 0.802 | 0.841 | 0.441 | 0.642 | 0.640 | 0.564 |
| | ProtSSN | 0.716 | 0.765 | 0.390 | – | – | – |
| Precision | ESM2 t30 | 0.824 | 0.802 | 0.510 | 0.648 | 0.644 | 0.496 |
| | ESM2 t33 | 0.851 | 0.795 | 0.566 | 0.661 | 0.634 | 0.530 |
| | ProtBert | 0.784 | 0.793 | 0.472 | 0.636 | 0.596 | 0.588 |
| | Ankh base | 0.846 | 0.817 | 0.499 | 0.674 | 0.646 | 0.494 |
| | GVP-GNN | 0.748 | 0.756 | 0.329 | 0.591 | 0.557 | 0.519 |
| | SaProt 35M | 0.821 | 0.783 | 0.485 | 0.632 | 0.615 | 0.548 |
| | SaProt 650M | 0.841 | 0.818 | 0.504 | 0.635 | 0.656 | 0.572 |
| | ProtSSN | 0.772 | 0.775 | 0.390 | – | – | – |
| Recall | ESM2 t30 | 0.775 | 0.731 | 0.432 | 0.433 | 0.423 | 0.360 |
| | ESM2 t33 | 0.748 | 0.861 | 0.384 | 0.467 | 0.478 | 0.367 |
| | ProtBert | 0.678 | 0.592 | 0.231 | 0.353 | 0.420 | 0.138 |
| | Ankh base | 0.789 | 0.831 | 0.303 | 0.467 | 0.490 | 0.280 |
| | GVP-GNN | 0.525 | 0.669 | 0.453 | 0.344 | 0.309 | 0.087 |
| | SaProt 35M | 0.582 | 0.954 | 0.411 | 0.322 | 0.840 | 0.349 |
| | SaProt 650M | 0.615 | 0.960 | 0.350 | 0.472 | 0.414 | 0.444 |
| | ProtSSN | 0.550 | 0.676 | 0.365 | – | – | – |
| F1-Negative | ESM2 t30 | 0.972 | 0.961 | 0.946 | 0.839 | 0.849 | 0.738 |
| | ESM2 t33 | 0.972 | 0.962 | 0.951 | 0.844 | 0.848 | 0.752 |
| | ProtBert | 0.963 | 0.952 | 0.945 | 0.834 | 0.837 | 0.779 |
| | Ankh base | 0.974 | 0.963 | 0.946 | 0.847 | 0.853 | 0.748 |
| | GVP-GNN | 0.954 | 0.953 | 0.924 | 0.836 | 0.837 | 0.774 |
| | SaProt 35M | 0.961 | 0.954 | 0.944 | 0.832 | 0.840 | 0.761 |
| | SaProt 650M | 0.964 | 0.960 | 0.946 | 0.837 | 0.850 | 0.765 |
| | ProtSSN | 0.956 | 0.954 | 0.944 | – | – | – |
| F1-Positive | ESM2 t30 | 0.799 | 0.765 | 0.467 | 0.519 | 0.510 | 0.417 |
| | ESM2 t33 | 0.796 | 0.774 | 0.457 | 0.547 | 0.545 | 0.433 |
| | ProtBert | 0.727 | 0.681 | 0.310 | 0.454 | 0.494 | 0.223 |
| | Ankh base | 0.817 | 0.779 | 0.377 | 0.552 | 0.557 | 0.357 |
| | GVP-GNN | 0.618 | 0.710 | 0.399 | 0.435 | 0.408 | 0.149 |
| | SaProt 35M | 0.681 | 0.709 | 0.445 | 0.427 | 0.462 | 0.427 |
| | SaProt 650M | 0.710 | 0.754 | 0.414 | 0.542 | 0.508 | 0.500 |
| | ProtSSN | 0.642 | 0.720 | 0.412 | – | – | – |
| Macro-F1 | ESM2 t30 | 0.885 | 0.863 | 0.707 | 0.679 | 0.680 | 0.578 |
| | ESM2 t33 | 0.884 | 0.868 | 0.704 | 0.696 | 0.697 | 0.593 |
| | ProtBert | 0.845 | 0.816 | 0.628 | 0.644 | 0.665 | 0.501 |
| | Ankh base | 0.895 | 0.871 | 0.662 | 0.700 | 0.745 | 0.552 |
| | GVP-GNN | 0.786 | 0.831 | 0.661 | 0.636 | 0.623 | 0.462 |
| | SaProt 35M | 0.821 | 0.832 | 0.695 | 0.629 | 0.651 | 0.594 |
| | SaProt 650M | 0.837 | 0.857 | 0.680 | 0.689 | 0.679 | 0.632 |
| | ProtSSN | 0.799 | 0.837 | 0.678 | – | – | – |

Table 22: Detailed fragment-level classification results on "MF50" split across **Act**, **BindI**, **Evo**, and **Motif** datasets. "MF50" refers to mixed-family splits with 50% sequence identity filtering applied at the fragment level. Metrics reported include Accuracy, Precision, Recall, Macro-F1, and Matthews Correlation Coefficient (MCC).

| Metric | Model | Act | BindI | Evo | Motif |
|---|---|---|---|---|---|
| Accuracy | ESM2 t30 | 0.819 | 0.937 | 0.853 | 0.884 |
| | ESM2 t33 | 0.814 | 0.934 | 0.841 | 0.906 |
| | ProtBert | 0.736 | 0.927 | 0.828 | 0.884 |
| | Ankh base | 0.824 | 0.920 | 0.866 | 0.901 |
| | GVP-GNN | 0.907 | 0.972 | 0.914 | 0.807 |
| | SaProt 35M | 0.928 | 0.976 | 0.939 | 0.901 |
| | SaProt 650M | 0.928 | 0.986 | 0.950 | 0.927 |
| | ProtSSN | 0.891 | 0.972 | 0.915 | 0.914 |
| Precision | ESM2 t30 | 0.659 | 0.834 | 0.681 | 0.458 |
| | ESM2 t33 | 0.603 | 0.755 | 0.682 | 0.547 |
| | ProtBert | 0.618 | 0.838 | 0.644 | 0.455 |
| | Ankh base | 0.661 | 0.733 | 0.727 | 0.508 |
| | GVP-GNN | 0.826 | 0.901 | 0.763 | 0.387 |
| | SaProt 35M | 0.810 | 0.943 | 0.857 | 0.509 |
| | SaProt 650M | 0.830 | 0.968 | 0.868 | 0.546 |
| | ProtSSN | 0.773 | 0.940 | 0.804 | 0.564 |
| Recall | ESM2 t30 | 0.670 | 0.819 | 0.684 | 0.461 |
| | ESM2 t33 | 0.634 | 0.775 | 0.682 | 0.543 |
| | ProtBert | 0.636 | 0.794 | 0.646 | 0.458 |
| | Ankh base | 0.665 | 0.732 | 0.729 | 0.501 |
| | GVP-GNN | 0.833 | 0.882 | 0.768 | 0.371 |
| | SaProt 35M | 0.823 | 0.929 | 0.858 | 0.505 |
| | SaProt 650M | 0.830 | 0.956 | 0.875 | 0.562 |
| | ProtSSN | 0.774 | 0.948 | 0.807 | 0.556 |
| Macro-F1 | ESM2 t30 | 0.647 | 0.809 | 0.667 | 0.457 |
| | ESM2 t33 | 0.605 | 0.753 | 0.669 | 0.542 |
| | ProtBert | 0.609 | 0.790 | 0.627 | 0.452 |
| | Ankh base | 0.647 | 0.718 | 0.716 | 0.499 |
| | GVP-GNN | 0.822 | 0.884 | 0.757 | 0.370 |
| | SaProt 35M | 0.807 | 0.931 | 0.849 | 0.504 |
| | SaProt 650M | 0.825 | 0.957 | 0.863 | 0.552 |
| | ProtSSN | 0.764 | 0.931 | 0.793 | 0.556 |
| MCC | ESM2 t30 | 0.815 | 0.926 | 0.852 | 0.875 |
| | ESM2 t33 | 0.810 | 0.922 | 0.840 | 0.898 |
| | ProtBert | 0.731 | 0.914 | 0.827 | 0.875 |
| | Ankh base | 0.821 | 0.906 | 0.865 | 0.892 |
| | GVP-GNN | 0.906 | 0.967 | 0.913 | 0.791 |
| | SaProt 35M | 0.926 | 0.971 | 0.938 | 0.892 |
| | SaProt 650M | 0.926 | 0.984 | 0.950 | 0.921 |
| | ProtSSN | 0.889 | 0.967 | 0.915 | 0.907 |

Table 23: Residue-level classification results on "MP30" split across **Act**, **BindI**, **Evo**, **Motif**, and **Dom** datasets. "MP30" refers to mixed-family splits with 30% sequence identity filtering applied at the full-sequence level.

| Metric | Model | Act | BindI | Evo | Motif | Dom |
|--------|-------|-----|-------|-----|-------|-----|
| | ESM2 t30 | 0.754 | 0.817 | 0.664 | 0.689 | 0.298 |
| | ESM2 t33 | 0.787 | 0.814 | 0.703 | 0.778 | 0.264 |
| | ProtBert | 0.593 | 0.667 | 0.542 | 0.586 | 0.338 |
| Recall | Ankh base | 0.794 | 0.901 | 0.777 | 0.787 | 0.389 |
| | Ankh large | 0.766 | 0.835 | 0.761 | 0.616 | 0.392 |
| | ProtT5 xl_uniref50 | 0.760 | 0.894 | 0.758 | 0.730 | 0.465 |

Table 24: Residue-level classification performance across **Act** and **Evo** datasets. "MP50" refers to mixed-family splits with 50% sequence identity filtering applied at full-sequence levels. Metric reported Recall.

| Model | Act | | | | Evo | | | |
|-------|------|------|------|------|------|------|------|------|
| | MP30 | MP50 | MP70 | MP90 | MP30 | MP50 | MP70 | MP90 |
| ESM2 t30 | 0.754 | 0.847 | 0.887 | 0.926 | 0.664 | 0.818 | 0.862 | 0.911 |
| ESM2 t33 | 0.787 | 0.870 | 0.892 | 0.894 | 0.703 | 0.843 | 0.884 | 0.932 |
| ProtBert | 0.593 | 0.781 | 0.810 | 0.857 | 0.542 | 0.677 | 0.823 | 0.866 |
| Ankh base | 0.794 | 0.886 | 0.892 | 0.901 | 0.777 | 0.864 | 0.909 | 0.938 |
| Ankh large | 0.766 | 0.862 | 0.853 | 0.863 | 0.761 | 0.846 | 0.902 | 0.926 |
| ProtT5 xl_uniref50 | 0.760 | 0.886 | 0.889 | 0.887 | 0.758 | 0.860 | 0.905 | 0.939 |

