# OpenReview forum: "VenusX: Unlocking Fine-Grained Functional Understanding of Proteins"
_ICLR.cc/2026/Conference — ICLR 2026 Poster_

### Official Review · Reviewer_7f4D · 2025-10-30

**Soundness:** 3
**Presentation:** 3
**Contribution:** 2
**Rating:** 6
**Confidence:** 4

**Summary:**

This study presents VenusX, a comprehensive benchmark for assessing protein representation learning models with respect to fine-grained, intra-protein functional understanding. VenusX encompasses three primary task categories: residue-level binary classification, fragment-level multi-class classification, and pairwise functional similarity scoring. The benchmark is built from over 878,000 curated samples sourced from InterPro, BioLiP, and SAbDab, featuring both mixed-family and cross-family data splits at multiple sequence identity thresholds to evaluate in-distribution and out-of-distribution generalization. The authors benchmark a diverse range of models—including sequence-only, sequence-structure, and structure-only approaches—across all tasks. Key observations reveal that sequence-based models excel in in-distribution performance, whereas sequence-structure hybrid models exhibit superior generalization to unseen families, with cross-family prediction remaining a significant challenge. The benchmark, along with its code, data, and leaderboard, is released as open-source.

**Strengths:**

- VenusX represents the first large-scale benchmark specifically targeting fine-grained, sub-protein functional understanding, addressing a critical gap in existing protein benchmarks that largely focus on protein-level properties.
- The benchmark encompasses a diverse set of tasks (residue-level, fragment-level, and pairwise functional similarity), multiple annotation types (active sites, binding sites, epitopes, etc.), and meticulously designed data splits (mixed-family, cross-family, and multiple sequence identity thresholds), enabling a thorough evaluation of model capabilities.
- The study conducts an extensive assessment of state-of-the-art models across different modalities (sequence-only, structure-only, and sequence-structure hybrids), providing valuable insights into their respective strengths and limitations.
- All code, data, and a leaderboard will be released as open-source, facilitating community adoption and future research. The methodology for data curation, task formulation, and evaluation is documented in sufficient detail to ensure reproducibility.

**Weaknesses:**

- Although a wide range of models is evaluated, the paper provides limited insight into why certain approaches—such as sequence-structure hybrids—perform better on cross-family splits. Incorporating ablation studies or attention visualization could shed light on the learned representations and their functional relevance.
- Performance on the epitope prediction (Epi) task is notably low (AUPR < 0.3), even for top-performing models. A more detailed discussion of the intrinsic challenges of this task, along with potential reasons for the poor results, would strengthen the analysis.
- The benchmark’s scale (e.g., pairwise similarity tasks involving 177 billion negative pairs) and reliance on large models (e.g., ESM2-3B, ProtT5) entail substantial computational resources for full evaluation. Addressing the computational burden and discussing accessibility for researchers with limited resources would be valuable.
- For residue-level tasks on the BioLiP (BindB) and SAbDab (Epi) datasets, only sequence-only models are evaluated (as shown in Table 4). Including structure-aware baselines where feasible would offer a more complete assessment of model performance.

**Questions:**

- Could the authors provide further analysis or intuition on why sequence-structure hybrid models, such as SaProt, exhibit superior generalization on cross-family splits compared to sequence-only models? Is this improvement primarily driven by the structural vocabulary, the training objective, or a combination of both factors?
- The epitope prediction task (Epi) reports very low AUPR scores. What are the specific challenges underlying this task—such as data quality, label definition, or the conformational diversity of epitopes—that contribute to its difficulty? What potential directions could be explored to improve performance in future work?
- For the pairwise functional similarity scoring task, evaluation is conducted on a fixed sample of 10,000 positive and negative pairs. Was the sensitivity of the AUC score to this particular random sample analyzed? Reporting the variance across different random samples could provide a clearer picture of evaluation robustness.
- The GVP-GNN model, a structure-only approach, is trained from scratch, whereas other models use frozen pretrained checkpoints. Could the authors elaborate on the rationale for this choice? Were experiments conducted using pretrained GVP-GNN models or treating GVP-GNN as a frozen feature extractor to enable a fairer comparison in terms of parameter efficiency and training data usage?
- What are the key innovations and contributions of VenusX compared to prior work such as ProteinBench [1]?

[1] Ye, F., Zheng, Z., Xue, D., Shen, Y., Wang, L., Ma, Y., … & Gu, Q. (2024). ProteinBench: A holistic evaluation of protein foundation models. arXiv preprint arXiv:2409.067.

---

> ### Author Response · Authors · 2025-11-22
>
> We sincerely thank the reviewer for their thorough summary and for recognizing the strengths of VenusX as the "first large-scale benchmark" for this critical task, highlighting our "diverse set of tasks" and "meticulously designed data splits". We are grateful for the constructive and detailed feedback. We will address each weakness and question in order.
>
> ## Weaknesses:
> ### W1. Limited Insight on Hybrid Models
> This is a key finding. Our hypothesis is that sequence-structure models excel in cross-family (OOD) settings because **protein structure is more conserved than sequence**. In scenarios where sequence homology is low (typical of remote homologs in unseen families), the sequence signal fades, and the 3D structural signal becomes the dominant indicator of function. We conducted new experiments on the MP30 split in **Table 15 (Appendix D.4)**, its low identity threshold simulates the "low-homology" challenge inherent in cross-family generalization.
>
> | Model | Res_Act_MP30 | Res_Act_MP50 | Res_Evo_MP30 | Res_Evo_MP50 |
> | :--- | :---: | :---: | :---: | :---: |
> | ESM2 t33 (Seq-only) | 0.787 | **0.870** | 0.703 | 0.861 |
> | SaProt 650M (Seq-Struct) | **0.792** | 0.850 | **0.714** | **0.866** |
>
> The sequence-only ESM2 performs comparably to the hybrid SaProt (0.870 vs. 0.850 on Act). However, on the harder MP30 split, **SaProt overtakes ESM2** (0.792 vs. 0.787 on Act; 0.714 vs. 0.703 on Evo). This demonstrates that SaProt successfully leverages structural information to maintain performance when sequence similarity drops. This "structural safety net" explains its superior robustness on cross-family tasks.
>
> ### W2. Low performance on Epi task
> We attribute the uniformly low AUPR (< 0.3) to three intrinsic challenges inherent to the task formulation:
> - **Conformational Complexity**: Epitopes are predominantly conformational (non-continuous), requiring high-order structural reasoning that current models struggle to capture effectively.
> - **Antibody-Agnosticism**: Unlike paratope-specific prediction, this task requires identifying inherently "bindable" surface regions without knowledge of the specific binding partner, which makes the objective significantly more difficult.
> - **Label Noise**: The ground truth relies on a geometric proxy (10Å cutoff from SAbDab), which introduces inherent noise compared to biological affinity data. This performance gap highlights a critical deficiency in current state-of-the-art models, underscoring the necessity of VenusX to drive progress in this complex domain.
>
> We have integrated this analysis into the Discussion section.
>
> > However, current models perform poorly on epitope prediction, revealing limitations in reasoning about conformational and antibody-independent features.
>
> ### W3. Computational Burden
> To ensure accessibility for researchers with limited resources, we implemented two key strategies:
> - **Hierarchical Data Splits**: We provide datasets at varying sequence identity thresholds (50%, 70%, 90%) and have additionally released **MP30 splits** for select large-scale tasks (e.g., Res_Act). These stricter filters result in significantly smaller datasets, enabling efficient prototyping and evaluation without processing the full-scale benchmark.
> - **Pre-computed Embeddings**: We will release pre-extracted embeddings for all baseline models upon acceptance. This eliminates the need for expensive inference on large foundation models, allowing researchers to focus on lightweight downstream head training.
>
>
> ### W4. Missing Structure-aware Baselines
> Thanks for your suggestion. The results for structure-aware models (e.g., SaProt, GVP-GNN) on the BindB and Epi tasks have been incorporated into **Table 4**.
> | Target | Split | SaProt-35M | SaProt-650M | ProtSSN | GVP-GNN |
> | :--- | :--- | :---: | :---: | :---: | :---: |
> | **BindB** | MF50 | 0.374 | 0.388 | - | 0.301 |
> | | MP70 | 0.330 | 0.389 | - | 0.337 |
> | | MP90 | 0.354 | 0.411 | - | 0.374 |
> | **Epi** | MF50 | 0.182 | 0.195 | 0.196 | 0.118 |
> | | MP70 | 0.194 | 0.237** | 0.200 | 0.139 |
> | | MP90 | 0.256 | 0.308 | 0.274 | 0.196 |

---

> ### Author Response · Authors · 2025-11-22
>
> ## Questions:
> ### Q1. Why Hybrid Models Generalize Better
> As addressed in our response to W1, we now provide new empirical evidence supporting our hypothesis. The superior performance of SaProt on the MP30 split demonstrates that its "improvement [is] primarily driven by... structural vocabulary" when sequence identity is low.
>
> We have expanded this analysis in the Discussion section.
>
> > We further show that sequence-structure models significantly outperform sequence-only baselines in low-identity settings (e.g., MP30), suggesting that structural priors are critical for generalization when sequence homology is weak (Appendix D.4).
>
>
> ### Q2. Specific Challenges of the Epi Task
> - **Specific Challenges**: As addressed in our response to W2, we identify the primary challenges of the Epi task as the 'antibody-independent' nature of the task, the "conformational diversity of epitopes", and the noisy label definition. We have added this discussion to the Discussion section.
> > However, current models perform poorly on epitope prediction, revealing limitations in reasoning about conformational and antibody-independent features.
> - **Potential Directions**: (1) As shown in Table 4, training more powerful hybrid PLMs may improve the performance (e.g., SaProt); (2) Considering a more fine-grained structural relationship modeling method between residues may improve the detection process (e.g., ProtSSN).
>
>
> ### Q3. Robustness of Pairwise Similarity Sampling
> As detailed in Section 3.4, we repeated the sampling procedure (10,000 positive and 10,000 negative pairs) three times using different random seeds. The results in **Table 6** are reported as 'mean(std. dev.)' (e.g., '96.1(0.1)'). The low standard deviation across all tasks confirms that our sampling strategy is highly robust.
>
> ### Q4. GVP-GNN Trained From Scratch
> This choice was driven by two factors:
> - **Standard Protocol & Fairness**: "Training from scratch" is the standard evaluation protocol for this class of geometric GNNs [1]. Crucially, this ensures fairness in terms of computational scale: the total trainable parameters of the GVP-GNN are comparable to the linear classification heads trained on top of the frozen PLMs.
>
> | Model | ProtSSN | SaProt 650M | GVP-GNN |
> | :--- | :---: | :---: | :---: |
> | Trainable Parameters | 1.641M | 1.641M  | 3.588M |
>
> - **Empirical Validation**: To address the suggestion regarding pre-trained structure models, we added an evaluation of GearNet (see **Table 15, Appendix D.2**). We observed that structural pre-training did not yield significant gains and, in some cases, exhibited lower generalization compared to the supervised GVP-GNN. This validates our usage of GVP-GNN as a robust, representative baseline.
>
> | Model | Res_Epi_MP50 | Res_Epi_MP70 | Res_Epi_MP90 |
> | :--- | :---: | :---: | :---: |
> | GVP-GNN | 0.118 | 0.139 | 0.196 |
> | GearNet | 0.116 | 0.129 | 0.167 |
>
> We have added the explanation in Sec 4.1.
>
> > This approach aligns with standard evaluation protocols for geometric GNNs [1] and ensures fairness in computational scale.
>
> [1] Jamasb A R, Morehead A, Joshi C K, et al. Evaluating representation learning on the protein structure universe. ICLR2024.
>
> ### Q5. Contribution Compared to ProteinBench
> ProteinBench is an excellent and contemporary work that was published on ICLR2025. ProteinBench and VenusX are highly complementary:
> - **Breadth vs. Depth**: ProteinBench is a "holistic" benchmark covering a very broad range of tasks, including "structure prediction" and "molecular dynamics". In contrast, VenusX provides a deep-dive into one critical, overlooked area: "fine-grained, intra-protein functional understanding".
> - **Our Key Innovation**: Our primary contribution is our "meticulously designed" cross-family OOD splitting methodology. As ProteinBench's own metareview notes, such "Specific details on dataset curation + splitting are extremely important". This rigorous focus on OOD generalization is a distinct contribution of VenusX.
>
> We have added a citation and discussion to our Related Work section to clarify our distinct contribution.
>
> > ProteinBench evaluates various methods for different tasks such as protein structure prediction, sequence design, structure design, sequence-structure design, and molecular dynamics.

---

### Official Review · Reviewer_WEHT · 2025-10-30

**Soundness:** 2
**Presentation:** 3
**Contribution:** 3
**Rating:** 2
**Confidence:** 4

**Summary:**

The paper curates data from a few protein databases to create some benchmarking tasks for protein function prediction. The tasks differ from most prior work in this space in that they focus on annotating sub-sequences of proteins corresponding to active sites, domains, etc. Given the evolutionary relationships between protein sequences, they exhibit clustering behavior, and thus non-trivial train test splits are important to evaluate models' extrapolation. They do a good job of guaranteeing distance between training and evaluation sequences.

The performance of variety of deep learning and alignment-based baselines are compared.

**Strengths:**

This problem is important for analyzing proteins, both for making predictions for the wide variety of uncharacterized natural proteins and for designing new proteins.

The benchmark appears to be well organized and accessible for future researchers.

**Weaknesses:**

I have key questions about how fragments are defined for certain annotation types. See below.

There are no non-deep-learning baselines for some tasks.

Both the fragment-level classification and fragment similarity tasks don't seem to be simulating a workflow that practitioners would encounter in practice. How practical is it to be presented with a pre-identified sub-sequence of the protein without knowledge of it's functional label? Usually, this is presented as a sequence segmentation problem, where the input is a full sequence and the output is a set of labeled segments.

The discussion of related work should have been in the main paper instead of Appendix.

When doing such a large exploration of baseline deep learning models, it is always hard to understand how much the results would have changed if more hyperparameter tuning had been done for each.

**Questions:**

I was surprised to see that fragment-level classification and pairwise similarity scoring tasks are done using localized annotation types such as active sites and binding sites. An enzyme has a small number of active site residues, and they are often non-consecutive. What is the corresponding 'fragment' for this task? In Fig 2, the fragments are fairly long. Did you just take the entire subsequence that includes the active site residues? I don't think this is justifiable as a prediction task, since there are no clear semantics to this subsequence. This issue might also serve as a confounder when comparing the alignment-based baselines on these tasks.

For the residue-level annotation task, I'm curious how an alignment-based technique would have worked (you do alignment between the query sequence and the training set). I expect that this would perform well for things like active site residues.

---

> ### Author Response · Authors · 2025-11-22
>
> We appreciate the reviewer's recognition of the benchmark's organization. We address the concerns below, focusing on clarifying the definition of our fragment-level tasks and providing the requested non-deep learning baselines.
>
> ## Weaknesses:
> ### W1. No Non-deep-learning Baselines
> We implemented the requested alignment-based baseline by sampling functional motifs from the training set as queries and searching against the test set using NCBI BLAST+ (`blastp-short`). To ensure a robust and fair comparison, we repeated the experiment **10 times** with different 100 random query samples of 100 InterPro IDs, and reported the performance on `Res_Act_MF50` and `Res_Act_MF70` of the **top-3 best runs** in **Table 18 (Appendix E)**. The detailed methodology and analysis have been added to Appendix E.
>
> | P-Identity Threshold | MF50 Run 1 | MF50 Run 2 | MF50 Run 3 | MF70 Run 1 | MF70 Run 2 | MF70 Run 3 |
> | :--- | :---: | :---: | :---: | :---: | :---: | :---: |
> | 100 (Exact Match) | 0.0407 | 0.0410 | 0.0417 | 0.0471 | 0.0480 | 0.0492 |
> | 90 (High Identity) | 0.0435 | 0.0431 | 0.0439 | 0.0497 | 0.0528 | 0.0520 |
>
> Even under this optimistic setting, BLAST performs extremely poorly; the best-case AUPR on `Act_MF50` is only **~0.043**, whereas deep learning models achieve **>0.85** (Table 4). This striking gap empirically demonstrates that fine-grained functional residues cannot be accurately predicted solely by sequence alignment.
>
> ### W2. Practicality of Fragment Tasks
> We respectfully disagree that this workflow is not practical. While sequence segmentation is indeed one valid approach (covered by our Residue-Level Task), our Fragment-Level Task models a **second, complementary** workflow designed to address specific machine learning challenges:
> - **Mitigating Multi-Label Bias**: Proteins frequently possess multiple functional domains, creating a "one-to-many" mapping problem where multi-label learning suffers from severe label bias and evaluation difficulties. Our fragment-level task avoids this bias by isolating specific functional units, allowing models to learn intrinsic functional signatures without the noise of global multi-label distributions.
> - **Simulating Stepwise Analysis**: This models a realistic two-stage pipeline where a researcher first identifies key residues (e.g., via our residue-level task) and then submits that specific fragment for fine-grained functional classification. This decouples localization from classification, forming a complete and robust analysis workflow..
>
> ### W3. Place of Related Work
> Thanks for your suggestion. We have moved the Related Work section from the Appendix to the main paper.
>
> ### W4. Hyperparameter Tuning
> To address the concern about result stability, we conducted additional experiments using three different random seeds (42, 3407, 12345) on two challenging datasets (`Res_Act_MP30, Res_BindI_MP30`)  in **Table 16 (Appendix D.3)**.
>
> | Model | Act_MP30 Mean | Act_MP30 Std | BindI_MP30 Mean | BindI_MP30 Std |
> | :--- | :---: | :---: | :---: | :---: |
> | ESM2 t30 | 0.765 | 0.011 | 0.818 | 0.003 |
> | ESM2 t33 | 0.785 | 0.004 | 0.826 | 0.012 |
> | ProtBert | 0.587 | 0.007 | 0.668 | 0.014 |
> | ProtT5 xl_uniref50 | 0.767 | 0.006 | 0.883 | 0.010 |
> | Ankh base | 0.798 | 0.010 | 0.895 | 0.008 |
> | Ankh large | 0.776 | 0.010 | 0.842 | 0.008 |
>
> As shown above, the performance variance is minimal. The standard deviation across all models is consistently low (typically **$\le$ 0.01**), and the relative ranking of models (e.g., Ankh > ESM2 > ProtBert) remains stable across seeds. This confirms that our reported results reflect the intrinsic quality of the representations rather than random fluctuations. We provide all scripts to facilitate further hyperparameter exploration by the community.

---

> > ### Comment · Reviewer_WEHT · 2025-11-25
> >
> > Can you clarify the generality of the 'Simulating Stepwise Analysis' approach? If I understand correctly, in this approach there would be no sequence segmentation procedure, but instead a residue-level classifier model would be used to detect certain key residues and then these would be used to define interesting fragments? Besides for active sites, where also would this be applicable?

---

> > > ### Author Response · Authors · 2025-11-25
> > >
> > > ### Clarification on the Generality of the Stepwise Workflow
> > > We thank the reviewer for the insightful follow-up. The reviewer's interpretation is correct: this "detect-then-classify" workflow is a fundamental paradigm in bioinformatics, often referred to as Hierarchical Annotation. This approach decouples localization (segmentation/detection) from characterization (fine-grained classification) and is applicable far beyond active sites.
> > >
> > > - **Universality in InterPro Standards**: This workflow strictly aligns with InterPro specifications (Appendix B.1). The [InterPro document](https://interpro-documentation.readthedocs.io/en/latest/faq.html) defines not only Active Sites but also Binding Sites, Motifs, and Domains as specific sequence signatures. Consequently, classifying these pre-identified continuous regions is the standard operational procedure across all these functional categories, rather than an exception.
> > > - **Consistency with Structural Paradigms**: This two-stage paradigm is central to structural databases like **CATH** [1] and **TED** [2]. Standard pipelines first decompose proteins into constituent domains via segmentation algorithms (localization), then treat these extracted domains as independent inputs for superfamily classification (characterization). Our Fragment-Level Task rigorously evaluates this second critical step: assessing whether a model can resolve fine-grained functional identity (e.g., distinguishing between >13,000 families) based on the local signal, a prerequisite for automated domain annotation.
> > >
> > > [1] Orengo C A, Michie A D, Jones S, et al. CATH–a hierarchic classification of protein domain structures[J]. Structure, 1997.
> > >
> > > [2] Lau A M, Bordin N, Kandathil S M, et al. Exploring structural diversity across the protein universe with The Encyclopedia of Domains[J]. Science, 2024.

---

> ### Author Response · Authors · 2025-11-22
>
> ## Questions:
> ### Q1. The "Fragment" Definition for Active Sites
> We agree that active sites are typically composed of non-consecutive residues. VenusX addresses this biological reality through a strict separation of task designs:
> - **Differentiation of Task Objectives**: We strictly distinguish between **Residue-Level Tasks** (Sec 3.1) and **Fragment-Level Tasks** (Sec 3.2). The former predicts labels for **individual residues** within the full sequence, including consecutive or non-consecutive sites. The latter classifies **continuous subsequences**, not artificially concatenated residues.
> - **Adherence to Bioinformatics Standards**: The fragment definitions are grounded in [InterPro document](https://interpro-documentation.readthedocs.io/en/latest/entries_info.html) methodology (also can be found at Appendix B.1), which relies on conserved 1D sequence motifs ("signatures") for functional annotation. While active sites function in 3D, their identification in large-scale sequence databases necessitates the detection of these continuous linear fingerprints. Our task benchmarks this specific, standard capability.
> - **Exclusion of 3D-Only Features**: Crucially, for annotation types defined solely by 3D spatial proximity without robust 1D conserved motifs (specifically **BioLiP** binding sites and **SAbDab** epitopes), we exclusively implemented residue-level tasks. We did not create fragment tasks for these targets, ensuring our experimental design aligns with the physical constraints of protein structures.
>
> We have added a more detailed explanation in Sec 3.2 to avoid misunderstanding.
>
> > Crucially, inputs for this task are continuous sequence motifs (e.g., signature sequences defined by InterPro). We do not artificially concatenate non-consecutive residues. Consequently, tasks defined solely by 3D spatial proximity without conserved 1D motifs (e.g., BioLiP binding sites, SAbDab epitopes) are excluded from this category and handled exclusively in residue-level tasks.
>
>
> ### Q2. Alignment-based Baselines
> As addressed in W2, we have added the BLAST experiments and corresponding analysis to Appendix E.

---

### Official Review · Reviewer_4KXz · 2025-10-31

**Soundness:** 3
**Presentation:** 3
**Contribution:** 3
**Rating:** 6
**Confidence:** 3

**Summary:**

This paper introduces VENUSX, a pioneering benchmark for assessing deep learning models’ fine-grained functional understanding of proteins, beyond whole-protein tasks. It highlights the limitations of current models in capturing localized mechanisms—crucial for enzyme engineering, annotation, and interpretability—by curating >878,000 samples from InterPro, BioLiP, and SAbDab. VENUSX features three task types: residue-level binary classification, fragment-level multi-class classification, and pairwise similarity scoring, with mix- and cross-family splits at 50/70/90% identity. Baselines (ESM2, SaProt, GVP-GNN, etc.) show global performance does not guarantee local accuracy. All code, data, and leaderboards are open-sourced.

**Strengths:**

1.	VENUSX introduces a new focus on fine-grained protein functional understanding through novel residue-, fragment-, and similarity-based tasks, with cross-family splits to test generalization, addressing a key gap in evaluating localized biological signals.

2.	The benchmark is well-constructed from over 878k curated samples, with detailed task definitions, appropriate metrics, diverse baselines, and clear documentation, making it practical and accessible for research.

**Weaknesses:**

1. The evaluation lacks newer protein language models such as ESM3, which models sequence, structure, and function jointly; its absence limits the benchmark’s ability to reflect current state-of-the-art performance on fine-grained functional tasks.

2. Residue- and fragment-level tasks use only frozen residue embeddings passed through two linear layers; the lack of experiments on fine-tuning language models leaves unaddressed whether supervised adaptation can improve capture of localized functional signals.

3. Many of the tasks require training new classification heads on top of each protein representation model, which may be computationally demanding and could limit the benchmark’s accessibility and widespread adoption.

**Questions:**

1. Given the diversity of annotation sources (InterPro, BioLiP, SAbDab) and task types, could the authors provide an empirical analysis (e.g., correlation of model rankings, overlap in difficult examples, or transfer learning performance) across the seven residue-level and five fragment-level subtasks?

2. The experimental setup uses only frozen encoders with linear probes for all protein language models. Could the authors explain whether fine-tuning was attempted and, if not, provide a rationale (e.e., computational constraints, risk of overfitting on smaller splits, or intent to test pretraining quality)?

3. Proteins are often truncated to meet the input length limitations of the models. How are the protein structures modified in these cases to ensure that each residue is correctly mapped to its spatial location in the sequence+structure models? Providing more details on this process would significantly strengthen confidence in the fairness and rigor of the evaluation.

4.  For many entries, the benchmark relies on predicted structures from the AlphaFold Database rather than experimentally determined structures from PDB. While this is necessary to achieve the benchmark's massive scale, it introduces a potential confounding variable. An analysis of model performance on high-confidence predicted structures versus experimental structures would be a valuable addition.

5. For the classification tasks, GVP-GNN is the primary structure-only model evaluated and is trained from scratch. Including other modern, pre-trained structure-only models in the comparison would provide a more complete picture of that model class.

---

> ### Author Response · Authors · 2025-11-22
>
> We sincerely thank the reviewer for their accurate summary and for recognizing the strengths of VenusX as a "pioneering" and "well-constructed" benchmark that addresses a "key gap" in the field. We will address each weakness and question in order.
>
> ## Weaknesses:
> ### W1. Lack of newer models like ESM3
> We appreciate the suggestion. Our decision to omit ESM3 was based on two key factors:
> - **Data Contamination Concerns**: ESM3 explicitly incorporates InterPro annotations into its pre-training. Evaluating it on VenusX would confound the assessment of representation generalization with training data memorization, creating an inequitable comparison against baselines that were not exposed to these labels.
> - **Practical Constraints**: The restrictive license by ESM3 prohibits using its outputs "*to train any other... model similar to EvolutionaryScale’s AI Model*", introduces ambiguity for its use in benchmark training pipelines, even for non-commercial research.
>
> ### W2. Lack of Fine-Tuning Experiments
> This point is directly related to Q2, so we will address them here together.
> - **Methodology**: As the reviewer correctly hypothesized ("intent to test pretraining quality"), our goal was to test the *intrinsic quality of the pre-trained representations* using the standard frozen encoder setup.
> - **Accessibility**: To ensure broad community accessibility, we avoided the prohibitive computational costs associated with full-model fine-tuning. For example, training linear probes on the 'Dom' dataset needs >7 days of GPU time.
>
> ### W3. Computational Demand & Accessibility
> As addressed in our response to **W2**, we have designed the benchmark with this in mind. The flexible data splits (50/70/90%) and our plan to release pre-computed embeddings (ESM2/Ankh/ProtBert) are direct solutions to ensure accessibility.

---

> ### Author Response · Authors · 2025-11-22
>
> ## Questions:
> ### Q1. Cross-Task Analysis
> We have performed this analysis and added it to the appendix **Figure 9 (Appendix C)**.
> - **Residue-Level (Figure 9a)**: We found strong positive correlations ($\rho$ between 0.79 and 0.90) between Act, BindI, and Evo tasks, confirming their functional relationship.
> - **Fragment-Level (Figure 9b)**: We found a striking zero correlation ($\rho = -0.01$) between the Act and Motif tasks. This empirically proves that classifying a general "Motif" is a fundamentally different challenge from classifying a specific "Active Site motif," strongly supporting our benchmark's design.
>
> ### Q2. Rationale for Frozen Encoders
> As addressed in our response to **W2**, this was a deliberate choice for methodological rigor (testing pre-training quality) and practical accessibility (avoiding prohibitive compute costs). We have clarified this rationale in Sec 4.1.
> > This protocol isolates the intrinsic quality of pre-trained representations from fine-tuning confounders and ensures computational accessibility on this large-scale benchmark.
>
> ### Q3. Structure Truncation for Structure Models
> We confirm that when a sequence is truncated (e.g., 1022 for residue-level tasks), we simultaneously **truncate the corresponding 3D structure file** (e.g., AlphaFold PDB) to include only the atoms belonging to those same residues. The residue-to-coordinate mapping is perfectly preserved, ensuring a "fair and rigorous" evaluation. We have added this explicit detail to Section 4.1.
> > For sequence-structure models, 3D structure files were truncated in parallel to include only the atoms of the retained residues, ensuring a perfect residue-to-coordinate mapping.

---

> ### Author Response · Authors · 2025-11-22
>
> ### Q4. Predicted vs. Experimental Structures
> The reviewer raises a valuable point. To quantify this, we conducted an ablation study on the Epitope (Epi) task, comparing performance using experimental **PDB structures** versus **AlphaFold2 (AF2) predictions** on the MP50 split.
>
> #### (a) Model Performance
> | Model | PDB | AF2 |
> | :--- | :---: | :---: |
> | SaProt 35M | 0.1820 | 0.1721 |
> | SaProt 650M | 0.1948 | 0.1889 |
> #### (b) AlphaFold pLDDT Statistics
> | Level | Mean | Min | Max | 25% | 50% |
> | :--- | :---: | :---: | :---: | :---: | :---: |
> | Protein | 91.01 | 42.00 | 98.71 | 89.10 | 92.70 |
> | Residue | 91.40 | 19.44 | 99.00 | 89.75 | 94.88 |
>
> We provide the performance comparison on PDB/AF2 structures and the pLDDT statistics of our dataset in the **Table 14 (Appendix D.1)**. The performance gap is minimal (e.g., for SaProt-650M: **0.1948 on PDB vs. 0.1889 on AF2 (Table 14(a))**). This consistency is explained by the high quality of the predicted structures in our dataset, which have a **mean pLDDT of 91.01 (Table 14(b)**). This confirms that using AF2 structures allows us to achieve the benchmark's massive scale without compromising evaluation validity.
>
> ### Q5. More Structure-Only Models
> We have added the evaluation of GearNet (a representative pre-trained structure model) on the Epitope tasks in **Table 15 (Appendix D.2)**.
>
> | Model | Res_Epi_MP50 | Res_Epi_MP70 | Res_Epi_MP90 |
> | :--- | :---: | :---: | :---: |
> | GVP-GNN | 0.118 | 0.139 | 0.196 |
> | GearNet | 0.116 | 0.129 | 0.167 |
>
> GearNet achieves AUPR scores comparable to the trained-from-scratch GVP-GNN (e.g., **0.116 vs 0.118** on MP50). This suggests that current structural pre-training objectives offer limited transfer gains over supervised geometric learning for this specific surface-based task. We have added more discussion to the revised Appendix D.1.

---

### Author Response · Authors · 2025-12-03
**Global Response to All Reviewers and Area Chair**

## Global Response to All Reviewers and Area Chair
We sincerely thank the reviewers (**4KXz**, **WEHT**, **7f4D**) for their constructive feedback and for recognizing VenusX as a "**pioneering**" and "**well-constructed**" benchmark that addresses a "**key gap**" in fine-grained protein understanding.
In response to the reviews, we have uploaded a revised manuscript with significant updates (highlighted in **red**). We have performed a set of new experiments and added multiple analysis sections to the Appendix. Below is a summary of the critical revisions.
### 1. **Resolution of Core Misunderstanding (Crucial for Validity)**

Reviewer **WEHT (W1, Q1, W3)** raised concerns regarding the definition and practicality of "Fragment-Level" tasks. We have clarified this fundamental ambiguity:

- **Definition**: Inputs are **continuous sequence motifs** (e.g., [InterPro signatures](https://interpro-documentation.readthedocs.io/en/latest/entries_info.html)), not artificially concatenated non-consecutive residues. Non-consecutive features are handled exclusively by our **Residue-Level** tasks.
- **Standard Paradigm**: As clarified in our detailed response (and new text in **Sec 3.2**), this "detect-then-classify" workflow is the standard paradigm for InterPro and structural databases like **CATH** and **TED**, which decompose proteins into modular units (domains/motifs) for fine-grained classification.
- **Empirical Validation (New Baseline, Appendix E)**: To address the concern that deep learning might be unnecessary (**WEHT W2**), we implemented a BLAST baseline. The results (AUPR $\approx$ 0.04 vs. DL models > 0.85) empirically prove that traditional alignment fails on these fine-grained tasks, validating the necessity of our benchmark.

### 2. **Major Experimental Updates (Data-Driven Rebuttal)**

We conducted extensive experiments to address specific technical queries:

- **Mechanism of Generalization (Appendix D.4)**: To explain why sequence-structure models generalize better (**7f4D W1**), we evaluated models on a stricter low-homology split (MP30). Results show hybrids (e.g., SaProt) outperform sequence-only models when sequence identity drops, confirming structural priors as the key to OOD generalization.
- **Structure-Only Model Evaluation (Appendix D.2)**: We added **GearNet (pre-trained)** to the comparison (**4KXz Q5**). Results indicate structural pre-training yields limited gains over supervised learning (GVP-GNN) for the surface-sensitive Epitope task.
- **Structure Quality Ablation (Appendix D.1)**: We compared performance using Predicted (AF2) vs. Experimental (PDB) structures (**4KXz Q4**). The minimal performance gap (< 0.01 AUPR) confirms the reliability of using high-confidence AF2 structures (mean pLDDT > 91) to achieve scale.
- **Missing Baselines Completed (Table 4)**: We completed the evaluation of structure-aware models (SaProt, GVP-GNN) on the BindB and Epi tasks (**7f4D W4**).
- **Cross-Task Correlation (Appendix C)**: We added a heatmap analysis revealing that while some functional tasks are correlated, others (like Motif vs. Act) require distinct modeling capabilities (**4KXz Q1**).
- **Task Difficulty Insight**: We attribute the uniformly low performance on the Epi task to intrinsic challenges (conformational, antibody-independent nature) rather than model failure, identifying this as a key future direction (**7f4D W2**).

### 3. **Scientific Rigor, Positioning & Accessibility**
- **Exclusion of ESM3**: We excluded ESM3 to avoid Data Contamination (**4KXz W1**). Since ESM3 utilizes InterPro in pre-training, evaluating it on VenusX would confound generalization with memorization, violating the benchmark's scientific integrity.
- **Protocol Rationale (Frozen vs. Fine-tuning)**: We maintained the "frozen encoder" protocol to (1) isolate the intrinsic quality of pre-trained representations and (2) ensure computational accessibility. Full fine-tuning would incur prohibitive costs (e.g., >7 days per task), limiting community adoption (**4KXz W2, 7f4D W3**).
- **Distinction from ProteinBench**: We clarified in Section 5 that while ProteinBench offers Breadth (holistic tasks), VenusX offers Depth (intra-protein function) and contributes a unique, rigorous Cross-Family (OOD) Splitting methodology essential for evaluating generalization (**7f4D Q5**).
- **Stability (Appendix D.3)**: We verified result stability across three random seeds, showing negligible variance (**WEHT W5**).

We believe these data-driven revisions conclusively address the reviewers' concerns and demonstrate the robustness of VenusX, and thank the efforts to community for the Reviewers and the Area Chair!

---

### Meta-Review · Area_Chair_UzsJ · 2025-12-26

**Summary:**

This paper proposes a large-scale benchmark for fine-grained intra-protein functional understanding (residue-, fragment-, and pairwise-similarity tasks) with carefully constructed mixed-family and cross-family splits at multiple sequence identity thresholds, and an extensive set of baseline models.

The reviewers generally recognized the value of the benchmark in addressing a key gap: moving beyond whole-protein classification to localized fine-grained functional understanding. Across reviews, the main issues were
- a core misunderstanding regarding the definition of "fragment-level" tasks
- the lack of non-deep-learning baselines
- questions about the scope and accessibility (training protocol, computational burden, exclusion of newer models such as ESM3)

The authors provided a comprehensive rebuttal, including clarifications on the fragment definition and positioning, new baselines (e.g., BLAST, GearNet, structure-aware models), new ablation studies (AF2 vs PDB), robustness analyses.

Overall, the remaining concerns are mostly about benchmark protocol choices and scope rather than validity, and I recommend accept.

**Reviewer Concerns:**

**Addressed concerns**
- The authors clarified that fragment-level inputs are continuous sequence motifs/signatures (as defined by InterPro), while non-contiguous sites are handled by the separate residue-level task. They positioned the workflow as a standard "detect-then-classify" paradigm.
- They added a BLAST baseline and showed very poor performance compared to DL methods, empirically supporting that the tasks are non-trivial.
- They included structure-aware baselines on the Bindb/Epi tasks and more structure-only models such as GearNet.
- They added an ablation study to compare AF2 and PDB structures, showing a small performance gap.
- They made several clarifications and robustness checks.

**Outstanding concerns**
- The authors provide a reasonable explanation on the exclusion of ESM3 (data contamination, licensing), but the absence still limits "current SOTA coverage" to some extent. This is a scope limitation rather than a critical flaw.
- The authors justify the frozen protocol for isolating representation quality and for accessibility (>7 days per task). While justified, the limitation that the benchmark does not currently benchmark fine-tuning performance remains.
- Performance on the Epi task remains very low. The authors attribute this to intrinsic difficulties and label noise. I think the task is still valuable, though the users should be cautioned in interpreting the results.

**Reviewer Scores:**

I expect reviewer WEHT to increase their score to 6. This reviewer's rejection was based on the fragment definition/practicality and missing non-DL baselines. The rebuttal addresses both issues.

---

### Decision · Program_Chairs · 2026-01-26

Accept (Poster)